# Staying up to Date with Online Content Changes Using Reinforcement Learning for Scheduling

**Andrey Kolobov**
Microsoft Research
Redmond, WA-98052
akolobov@microsoft.com

**Yuval Peres**
yperes@gmail.com

**Cheng Lu**
Microsoft Bing
Bellevue, WA-98004
Cheng.Lu@microsoft.com

**Eric Horvitz**
Microsoft Research
Redmond, WA-98052
horvitz@microsoft.com

## Abstract

From traditional Web search engines to virtual assistants and Web accelerators, services that rely on online information need to continually keep track of remote content changes by explicitly requesting content updates from remote sources (e.g., web pages). We propose a novel optimization objective for this setting that has several practically desirable properties, and efficient algorithms for it with optimality guarantees even in the face of mixed content change observability and initially unknown change model parameters. Experiments on 18.5M URLs crawled daily for 14 weeks show significant advantages of this approach over prior art.

## 1 Introduction

As the Web becomes more and more dynamic, services that rely on web data face the increasingly challenging problem of keeping up with online content changes. Whether it be a continuous-query system [26], a virtual assistant like Cortana or Google Now, or an Internet search engine, such a service tracks many remote sources of information – web pages or data streams [27]. Users expect these services, which we call *trackers*, to surface the latest information from the sources. This is easy if sources *push* content updates to the tracker, but few sources do. Instead, major trackers such as search engines must continually decide when to re-*pull* (*crawl*) data from sources to pick up the changes. A policy that makes these decisions well solves the *freshness crawl scheduling problem*.

Freshness crawl scheduling has several challenging aspects. For most sources, the tracker finds out whether they have changed only when it crawls them. To guess when the changes happen, and hence should be downloaded, the tracker needs a predictive model whose parameters are initially unknown. Thus, the tracker needs to learn these models *and* optimize a freshness-related objective when scheduling crawls. For some web pages, however, sitemap polling and other means can provide trustworthy near-instantaneous signals that the page has changed in *a* meaningful way, though not what the change is exactly. But even with these remote change observations and known change model parameters, freshness crawl scheduling remains highly nontrivial because the tracker cannot react to every individual predicted or actual change. The tracker's infrastructure imposes a *bandwidth constraint* on the average daily number of crawls, usually just a fraction of the change event volume. Last but not least, Google and Bing track many billions of pages [32] with vastly different importance and change frequency characteristics. The sheer size of this constrained learning and optimization problem makes low-polynomial algorithms for it a must, despite the availability of big-data platforms.

This paper presents a holistic approach to freshness crawl scheduling that handles all of the above aspects in a computationally efficient manner with optimality guarantees using a type of reinforcement

learning (RL) [29]. This problem has been studied extensively from different angles, as described in the Related Work section. The scheduling aspect per se, under various objectives and assuming known model parameters, has been the focus of many papers, e.g., [2, 10, 13, 25, 35]. In distinction from these works, our approach has *all* of the following properties: (i) optimality; (ii) computational efficiency; (iii) guarantee that every source changing at a non-zero rate will be occasionally crawled; (iv) ability to take advantage of remote change observations, if available. No other work has (iv), and only [35] has (i)-(iii). Moreover, learning change models previously received attention [11] purely as a preprocessing step. Our RL approach integrates it with scheduling, with convergence guarantees.

Specifically, our contributions are: (1) A natural freshness optimization objective based on harmonic numbers, and analysis showing how its mathematical properties enable efficient optimal scheduling. (2) Efficient optimization procedures for this bandwidth-constrained objective under complete, mixed, and lacking remote change observability. (3) A reinforcement learning algorithm that integrates these approaches with model estimation of [11] and converges to the optimal policy, lifting the known-parameter assumption. (4) An approximate crawl scheduling algorithm that requires learning far fewer parameters, and identifying a condition under which its solution is optimal.

## 2  Problem formalization

In settings we consider, a service we call *tracker* monitors a set $W$ of information sources. A source $w \in W$ can be a web page, a data stream, a file, etc, whose content occasionally changes. To pick up changes from a source, the tracker needs to *crawl* it, i.e., download its content. When source $w$ has changes the tracker hasn't picked up, the tracker is *stale* w.r.t. $w$; otherwise, it is *fresh* w.r.t. $w$. We assume near-instantaneous crawl operations, and a fixed set of sources $W$. Growing $W$ to improve *information completeness* [27] is also an important but distinct problem; we do not consider it here.

**Discrete page changes.** We define a *content change* at a source as an alteration at least minimally important to the tracker. In practice, trackers compute a source's content *digest* using data extractors, *shingles* [4], or *similarity hashes* [7], and consider content changed when its digest changes.

**Models of change process and importance.** We model each source $w \in W$'s changes as a Poisson process with *change rate* $\Delta_w$. Many prior works adopted it for web pages [2, 8, 9, 10, 11, 12, 35] as a good balance between fidelity and computational convenience. We also associate an *importance* score $\mu_w$ with each source, and denote these parameters jointly as $\vec{\mu}$. Importance score $\mu_w$ can be thought of as characterizing the time-homogeneous Poisson rate at which the page is served in response to the query stream, although in general it can be any positive weight measuring source significance [2]. *While scores $\mu_w$ are defined by, and known to, the tracker, change rates $\Delta_w$ need to be learned.*

**Change observability.** For most sources, the tracker can find out whether the source has changed only by crawling it. In this case, even crawling doesn't tell the tracker how many times the source has changed since the last crawl. We denote the set of these sources as $W^-$ and say that the tracker receives *incomplete change observations* about them. However, for other sources, which we denote as $W^o$, the tracker may receive near-instant notification whenever they change, i.e., get *complete remote change observations*. E.g., for web pages these signals may be available from browser telemetry or sitemaps. Thus the tracker's set of sources can be represented as $W = W^o \cup W^-$ and $W^o \cap W^- = 0$.

**Bandwidth constraints.** Even if the tracker receives complete change observations, it generally cannot afford to do a crawl upon each of them. The tracker's network infrastructure and considerations of respect to other Internet users limit its *crawl rate* (the average number of requests per day); the total *change rate* of tracked sources may be much higher. We call this limit *bandwidth constraint $R$*.

**Optimizing freshness.** The tracker operates in continuous time and starts fresh w.r.t. all sources. Our scheduling problem's solution is a policy $\pi$ — a rule that at every instant $t$ chooses (potentially stochastically) a source to crawl or decides that none should be crawled. Executing $\pi$ produces a *crawl sequence* of time-source pairs $CrSeq = (t_1, w_1), (t_2, w_2), \ldots$, denoted $CrSeq_w = (t_1, w), (t_2, w), \ldots$ for a specific source $w$. Similarly, the (Poisson) change process at the sources generates a change sequence $ChSeq = (t'_1, w'_1), (t'_2, w'_2), \ldots$, where $t'_i$ is a change time of source $w'_i$; its restriction to source $w$ is $ChSeq_w$. We denote the joint process governing changes at all sources as $P(\vec{\Delta})$.

## 3  Minimizing harmonic staleness penalty

We view maximizing freshness as minimizing costs the tracker incurs for the lack thereof, and associate the *time-averaged expected staleness penalty $J^\pi$* with every scheduling policy $\pi$:

$$J^\pi = \lim_{T \to \infty} \mathbb{E}_{\substack{CrSeq \sim \pi, \\ ChSeq \sim P(\vec{\Delta})}} \left[ \frac{1}{T} \int_0^T \sum_{w \in W} \mu_w C(N_w(t)) dt \right] \tag{1}$$

Here, $T$ is a planning horizon, $N_w(t)$ is the number of uncrawled changes source $w$ has accumulated by time $t$, and $C : \mathbb{Z}^+ \to \mathbb{R}^+$ is a *penalty function*, to be chosen later, that assigns a cost to every possible number of uncrawled changes. Note that $N_w(t)$ implicitly depends on the most recent time $w$ was crawled as well as on change sequence $ChSeq$, so the expectation is both over possible change sequences *and* possible crawl sequences $CrSeq$ generatable by $\pi$. Minimizing staleness means finding $\pi^* = \text{argmin}_{\pi \in \Pi} J^\pi$ under bandwidth constraint, where $\Pi$ is a suitably chosen policy class.

Choosing $C(n)$ that is efficient to optimize *and* induces "well-behaving" policies is of utmost importance. E.g., $C(n) = \mathbb{1}_{n>0}$, which imposes a fixed penalty if a source has any changes since last crawl [2, 10], can be optimized efficiently in $O(|W| \log(|W|))$ time over the class $\Pi$ of policies that crawl each source $w$ according to a Poisson process with a source-specific rate $\rho_w$. However, for many sources, the optimal $\rho_w^*$ is 0 under this $C(n)$ [2]. This is unacceptable in practice, as it leaves the tracker stale w.r.t. some sources forever, raising a question: why monitor these sources at all?

In this paper, we propose and analyze the following penalty:

$$C(n) = H(n) = \sum_{i=1}^n \frac{1}{i} \text{ if } n > 0, \text{ and } 0 \text{ if } n = 0 \tag{2}$$

$H(n)$ for $n > 0$ is the $n$-th harmonic number and has several desirable properties as staleness penalty:

***It is strictly monotonically increasing.*** Thus, it penalizes the tracker for every change that happened at a source since the previous crawl, not just the first one as in [10].

***It is discrete-concave, providing diminishing penalties:*** intuitively, while all undownloaded changes at a source matter, the first one matters most, as it marks the transition from freshness to staleness.

***"Good" policies w.r.t. this objective don't starve any source as long as that source changes.*** This is because, as it turns out, policies that ignore changing sources incur $J^\pi = \infty$ if $C(n)$ is as in Eq. 2 (see Prop. 1 in Section 4). In fact, this paper's optimality results and high-level approaches are valid for any concave $C(n) \geq 0$ s.t. $\lim_{n \to \infty} C(n) = \infty$, though possibly at a higher computational cost.

***It allows for efficiently finding optimal policies under practical policy classes.*** Indeed, $C(n) = H(n)$ isn't the only penalty function satisfying the above properties. For instance, $C(n) = n^d$ for $0 < d < 1$ and $C(n) = \log_d(1 + n)$ for $d > 1$ behave similarly, but result in much more computationally expensive optimization problems, as do other alternatives we have considered.

## 4 Optimization under known change process

We now derive procedures for optimizing Eq. 1 with $C(n) = H(n)$ (Eq. 2) under the bandwidth constraint for sources with incomplete and complete change observations, assuming that we know the change process parameters $\vec{\Delta}$ exactly. In Section 5 we will lift the known-parameters assumption. We assume $\vec{\mu}, \vec{\Delta} > 0$, because sources that are unimportant or never change don't need to be crawled.

### 4.1 Case of incomplete change observations

When the tracker can find out about changes at a source only by crawling it, we consider randomized policies that sample crawl times for each source $w$ from a Poisson process with rate $\rho_w$:

$$\Pi^- = \{CrSeq_w \sim Poisson(\rho_w) \, \forall w \in W^- | \vec{\rho} \geq 0\} \tag{3}$$

This policy class reflects the intuition that, since each source changes according to a Poisson process, i.e., roughly periodically, it should also be crawled roughly periodically. In fact, as Azar et al. [2] show, any $\pi \in \Pi^-$ can be de-randomized into a deterministic policy that is approximately periodic for each $w$. Since every $\pi \in \Pi^-$ is fully determined by the corresponding vector $\vec{\rho}$, we can easily express a bandwidth constraint on $\pi \in \Pi^-$ as $\sum_{w \in W^-} \rho_w = R$.

To optimize over $\Pi^-$, we first express policy cost (Eq. 1) in terms of $\Pi^-$'s policy parameters $\vec{\rho} \geq 0$:

**Proposition 1.** *For $\pi \in \Pi^-$, $J^\pi$ from Eq. 1 is equivalent to*

$$J^\pi = - \sum_{w \in W^-} \mu_w \ln \left( \frac{\rho_w}{\Delta_w + \rho_w} \right) \tag{4}$$

*Proof.* See the Supplement. Note that $J^\pi = \infty$ if $\rho_w = 0$ for any $w \in W^-$. The proof relies on properties of Poisson processes, particularly memorylessness. ∎

Thus, finding $\pi^* \in \Pi^-$ can be formalized as follows:

**Problem 1.** *[Finding $\pi^* \in \Pi^-$]*

INPUT: *bandwidth $R > 0$; positive importance and change rate vectors $\vec{\mu}, \vec{\Delta} > 0$.*

OUTPUT: *Crawl rates $\vec{\rho} = (\rho_w)_{w \in W^-}$ maximizing $\overline{J}^\pi = -J^\pi = \sum_{w \in W^-} \mu_w \ln \left( \frac{\rho_w}{\Delta_w + \rho_w} \right)$ subject to $\sum_{w \in W^-} \rho_w = R, \rho_w \geq 0$ for all $w \in W^-$.*

The next result readily identifies the optimal solution to this problem:

**Proposition 2.** *For $\vec{\mu}, \vec{\Delta} > 0$, policy $\pi^* \in \Pi^-$ parameterized by $\vec{\rho}^* > 0$ that satisfies the following equation system is unique, minimizes harmonic penalty $J^\pi$ in Eq. 2, and is therefore optimal in $\Pi^-$:*

$$\begin{cases} \rho_w = \frac{\sqrt{\Delta_w^2 + \frac{4\mu_w \Delta_w}{\lambda}} - \Delta_w}{2}, \text{for all } w \in W^- \\ \sum_{w \in W^-} \rho_w = R \end{cases} \tag{5}$$

---

**Algorithm 1:** LAMBDACRAWL-INCOMLOBS: finding the optimal crawl scheduling policy $\pi^* \in \Pi^-$ under incomplete change observations (Problem 1)

**Input:** $R \geq 0$ – bandwidth;
$\quad\quad \vec{\mu} > 0, \vec{\Delta} > 0$ – importance and change rates;
$\quad\quad \epsilon > 0$ – desired precision on $\lambda$
**Output:** $\vec{\rho}$ – vector of crawl rates for each source.

1 $\lambda_{lower} \leftarrow \frac{|W^-|^2 \min_{w \in W^- }\{\Delta_w\} \min_{w \in W^-}\{\mu_w\}}{|W^-| \max_{w \in W^-}\{\Delta_w\} R + R^2}$

2 $\lambda_{upper} \leftarrow \frac{|W^-|^2 \max_{w \in W^- }\{\Delta_w\} \max_{w \in W^-}\{\mu_w\}}{|W^-| \min_{w \in W^-}\{\Delta_w\} R + R^2}$

3 $\lambda \leftarrow$ BisectionSearch$(\lambda_{lower}, \lambda_{upper}, \epsilon)$

4 // see, e.g., Burden & Faires [6]

5 **foreach** $w \in W^-$ **do** $\rho_w \leftarrow \frac{-\Delta_w + \sqrt{\Delta_w^2 + \frac{4\mu_w \Delta_w}{\lambda}}}{2}$

6 Return $\vec{\rho}$

---

*Proof.* See the Supplement. The main insight is that for any $\vec{\mu}, \vec{\Delta} > 0$ the Lagrange multiplier method, which gives rise to Eq. system 5, identifies the only maximizer of $\overline{J}^\pi = -J^\pi$ (Eq. 4) with $\vec{\rho} > 0$, which thus must correspond to $\pi^* \in \Pi^-$. Crucially, that solution always has $\lambda > 0$. ∎

Eq. system 5 is non-linear, but the r.h.s. of Eqs. involving $\lambda$ monotonically decreases in $\lambda > 0$, so, e.g., bisection search [6] on $\lambda > 0$ can find $\vec{\rho}^*$ as in Algorithm 1.

**Proposition 3.** LAMBDACRAWL-INCOMLOBS *(Algorithm 1) finds an $\epsilon$-approximation to Problem 1's optimal solution in time $O(\log_2(\frac{\lambda_{upper} - \lambda_{lower}}{\epsilon})|W^-|)$.*

*Proof.* See the Supplement. The key step is showing that the solution $\lambda$ is in $[\lambda_{lower}, \lambda_{upper}]$. ∎

Note that a convex problem like this could also be handled using interior-point methods, but the most suitable ones have higher, cubic per-iteration complexity [5].

### 4.2 Case of complete change observations

If the tracker receives a notification every time a source changes, the policy class $\Pi^-$ in Eq. 3 is clearly suboptimal, because it ignores these observations. At the same time, crawling every source on every change signal is unviable, because the total change rate of all sources $\sum_{w \in W^o} \Delta_w$ can easily exceed bandwidth $R$. These extremes suggest a policy class whose members trigger crawls for only a fraction of the observations, dictated by a source-specific probability $p_w$:

$$\Pi^o = \{\text{for all } w \in W^o, \text{on each observation } o_w \text{ crawl } w \text{ with probability } p_w \,|\, 0 \leq \vec{p} \leq 1\} \tag{6}$$

As with $\Pi^-$, to find $\pi^* \in \Pi^o$ we first express $J^\pi$ from Eq. 1 in terms of $\Pi^o$'s policy parameters $\vec{p}$:

**Proposition 4.** *For $\pi \in \Pi^o$, $J^\pi$ from Eq. 1 is equivalent to $J^\pi = -\sum_{w \in W^o} \mu_w \ln(p_w)$ if $\vec{p} > 0$ and $J^\pi = \infty$ if $p_w = 0$ for any $w \in W^o$.*

*Proof.* See the Supplement. The key insight is that under any $\pi \in \Pi^o$, the number of $w$'s uncrawled changes at time $t$ is geometrically distributed with parameter $p_w$. ∎

Under any $\pi \in \Pi^o$, the crawl rate $\rho_w$ of any source is related to its change rate $\Delta_w$: every time $w$ changes we get an observation and crawl $w$ with probability $p_w$. Thus, $\rho_w = p_w \Delta_w$. Also, bandwidth $R > \sum_{w \in W^o} \Delta_w$ isn't sensible, because with complete change observations the tracker doesn't benefit from more crawls than there are changes. Thus, we frame finding $\pi^* \in \Pi^o$ as follows:

**Problem 2.** *[Finding $\pi^* \in \Pi^o$]*

INPUT: *bandwidth $R$ s.t. $0 < R \leq \sum_{w \in W^o}$; importance and change rate vectors $\vec{\mu}, \vec{\Delta} > 0$.*

OUTPUT: *Crawl probabilities $\vec{p} = (p_w)_{w \in W^o}$ subject to $\sum_{w \in W^o} p_w \Delta_w = R$ and $0 \leq p_w \leq 1$ for all $w \in W^o$, maximizing $\overline{J}^\pi = -J^\pi = \sum_{w \in W^o} \mu_w \ln(p_w)$.*

Non-linear optimization under inequality constraints could generally take exponential time in the constraint number. Our main result in this subsection is a *polynomial* optimal algorithm for Problem 2.

First, consider a relaxation of Problem 2 that ignores the inequality constraints:

**Proposition 5.** *The optimal solution $\vec{\hat{p}}^*$ to the relaxation of Problem 2 that ignores inequality constraints is unique and assigns $\hat{p}_w^* = \frac{R\mu_w}{\Delta_w \sum_{w' \in W^o} \mu_{w'}}$ for all $w \in W^o$.*

*Proof.* See the Supplement. The proof applies Lagrange multipliers. ∎

Our algorithm LAMBDACRAWL-COMPLOBS (Algorithm 2)'s high-level approach is to iteratively (lines 4-14) solve Problem 2's relaxations as in Prop. 5 (lines 5-6), each time detecting sources that *activate* (either meet or exceed) the $p_w \leq 1$ constraints (line 9). (Note that the relaxed solution never has $\hat{p}_w^* \leq 0$.) Our key insight, which we prove in the Supplement, is that any such source has $p_w^* = 1$. Therefore, we set $p_w^* = 1$ for each of them, adjust the overall bandwidth constraint for the remaining sources to $R_{rem} = R - p_w^* \Delta_w = R - \Delta_w$, and remove $w$ from further consideration (lines 10-12). Eventually, we arrive at a (possibly empty) set of sources for which Prop. 5's solution obeys all constraints under the remaining bandwidth (lines 15-16). Since Prop. 5's solution is optimal in this base case, the overall algorithm is optimal too.

---

**Algorithm 2:** LAMBDACRAWL-COMPLOBS: finding the optimal crawl scheduling policy $\pi^* \in \Pi^o$ under complete change observations (Problem 2)

---

1   LAMBDACRAWL-COMPLOBS:

    **Input:** $\vec{\mu}, \vec{\Delta}$ – importance and change rate vectors
2         $R$ s.t. $0 \leq R \leq \sum_{w \in W} \Delta_w$ – bandwidth;

    **Output:** $\vec{p}^*$ – vector specifying optimal per-page crawl probabilities upon receiving a change observation.

3   $W_{rem} \leftarrow W^o$     // remaining sources to consider
4   **while** $W_{rem}^o \neq \emptyset$ **do**
5      **foreach** $w \in W_{rem}^o$ **do**
6        $\hat{p}_w^* \leftarrow \frac{R\mu_w}{\Delta_w \sum_{w' \in W_{rem}^o} \mu_{w'}}$ for all $w \in W_{rem}^o$
7      $ViolationDetected \leftarrow False$
8      **foreach** $w \in W_{rem}^o$ **do**
9        **if** $\hat{p}_w^* \geq 1$ **then**
10          $p_w^* \leftarrow 1$
11          $R \leftarrow R - \Delta_w$    // reduce remaining bandwidth
12          $W_{rem}^o \leftarrow W_{rem}^o \setminus \{w\}$    // ignore $w$ onwards
13          $ViolationDetected = True$
14      **if** $ViolationDetected == False$ **then** break
15   **foreach** $w \in W_{rem}^o$ **do**
16      $p_w^* \leftarrow \hat{p}_w^*$
17   Return $\vec{p}^* = (p_w^*)_{w \in W^o}$

---

**Proposition 6.** LAMBDACRAWL-COMPLOBS *is optimal for Problem 2 and runs in time $O(|W^o|^2)$.*

*Proof.* See the Supplement. The proof critically relies on the concavity of $\overline{J}^\pi$. ∎

The $O(|W^o|^2)$ bound is loose. Each iteration usually discovers several active constraints at once, and for many sources the constraint is never activated, so the actual running time is close to $O(|W^o|)$.

### 4.3   Crawl scheduling under mixed observability

In practice, trackers have to simultaneously handle sources with and without complete change data under a *common* bandwidth budget $R$. Consider a policy class that combines $\Pi^-$ and $\Pi^o$:

$$\Pi^\ominus = \begin{cases} \text{For all } w \in W^-: \{CrSeq_w \sim Poisson(\rho_w) | \vec{\rho}\}, \\ \text{For all } w \in W^o: \{\text{on each change observation } o_w, \text{ crawl } w \text{ with probability } p_w | \vec{p}\} \end{cases} \quad (7)$$

For $\pi \in \Pi^\ominus$, Prop.s 1 and 4 imply that $J^\pi$ from Eq. 1 is equivalent to

$$J^\pi = -\sum_{w \in W^-} \mu_w \ln\left(\frac{\rho_w}{\Delta_w + \rho_w}\right) - \sum_{w \in W^o} \mu_w \ln(p_w) \quad (8)$$

Optimization over $\pi \in \Pi^{\ominus}$ can be stated as follows:

**Problem 3.** *[Finding $\pi^* \in \Pi^{\ominus}$]*

INPUT: *bandwidth $R > 0$; importance and change rate vectors $\vec{\mu}, \vec{\Delta} > 0$.*

OUTPUT: *Crawl rates $\vec{\rho} = (\rho_w)_{w \in W^-}$ and crawl probabilities $\vec{p} = (p_w)_{w \in W^\circ}$ maximizing*

$$\overline{J}^\pi = -J^\pi = \sum_{w \in W^-} \mu_w \ln\left(\frac{\rho_w}{\Delta_w + \rho_w}\right) + \sum_{w \in W^\circ} \mu_w \ln(p_w) \qquad (9)$$

*subj. to $\sum_{w \in W^-} \rho_w + \sum_{w \in W^\circ} p_w \Delta_w = R, \rho_w > 0$ for all $w \in W^-, 0 < p_w \le 1$ for all $w \in W^\circ$.*

The optimization objective (Eq. 9) is strictly concave as a sum of concave functions over the constrained region, and therefore has a unique maximizer. Finding it amounts to deciding how to split the total bandwidth $R$ into $R^o$ for sources with complete change observations and $R^- = R - R^o$ for the rest. For any candidate split, LAMBDACRAWL-COMPLOBS and LAMBDACRAWL-INCOMLOBS give us the reward-maximizing policy parameters $\vec{p}^*(R^o)$ and $\vec{\rho}^*(R^-)$, respectively, and Eq. 9 then tells us the overall value $\overline{J}^*(R^o, R^-)$ of that split. We also know that for the optimal split, $R^{o^*} \in [0, \min\{R, \sum_{w \in W_o} \Delta_w\}]$, as discussed immediately before Problem 2. Thus, we can find Problem 3's maximizer to any desired precision using a method such as Golden-section search [20] on $R^o$. LAMBDACRAWL (Algorithm 3) implements this idea, where SPLIT-EVAL-$\overline{J}^*$ (line 7) evaluates $\overline{J}^*(R^o, R^-)$ and OptMaxSearch denotes an optimal search method.

---

**Algorithm 3:** LAMBDACRAWL: finding optimal mixed-observability policy $\pi^* \in \Pi^{\ominus}$ (Problem 3)

**Input:** $R > 0$ – bandwidth;
  $\vec{\mu} > 0, \vec{\Delta} > 0$ – importance and change rates;
  $\epsilon^{\text{no-obs}}, \epsilon > 0$ – desired precisions
**Output:** $\vec{\rho}^*, \vec{p}^*$ – crawl rates and probabilities for sources without and with complete change observations.

1   $R^o_{min} \leftarrow 0$
2   $R^o_{max} \leftarrow \min\{R, \sum_{w \in W_o} \Delta_w\}$
3   $\vec{\rho}^*, \vec{p}^* \leftarrow$
    *OptMaxSearch*(Split-Eval-$\overline{J}^*$, $R^o_{min}, R^o_{max}, \epsilon$)
4   // E.g., Golden section search [20]
5   Return $\vec{\rho}^*, \vec{p}^*$
6
7   SPLIT-EVAL-$\overline{J}^*$:
**Input:** $R^o$ – bandwidth for sources with complete change observations, $R, \vec{\mu}, \vec{\Delta}, \epsilon^{\text{no-obs}}$
**Output:** $\overline{J}^*$ (Eq. 9) for the given split
8   $\vec{\rho} \leftarrow$ LAMBDACRAWL-INCOMLOBS($R - R^o, \vec{\mu}_{W^-}, \vec{\Delta}_{W^-}, \epsilon^{\text{no-obs}}$)
9   $\vec{p} \leftarrow$ LAMBDACRAWL-COMPLOBS($R^o, \vec{\mu}_{W^\circ}, \vec{\Delta}_{W^\circ}$)
10   Return
$$\sum_{w \in W^-} \mu_w \ln\left(\frac{\rho_w}{\Delta_w + \rho_w}\right) + \sum_{w \in W^\circ} \mu_w \ln(p_w)$$

---

**Proposition 7.** LAMBDACRAWL *(Algorithm 3) finds an $\epsilon$-approximation to Problem 3's optimal solution using $O(\log(\frac{R}{\epsilon}))$ calls to* LAMBDACRAWL-INCOMLOBS *and* LAMBDACRAWL-COMPLOBS.

*Proof.* This follows directly from the optimality of LAMBDACRAWL-INCOMLOBS and LAMBDACRAWL-COMPLOBS (Prop.s 2 and 6), as well as of OptMaxSearch such as Golden section, which makes $O(\log(\frac{R}{\epsilon}))$ iterations. ∎

## 5 Reinforcement learning for scheduling

All our algorithms so far assume known change rates, but in reality change rates are usually unavailable and vary with time, requiring constant re-learning. In this section we modify LAMBDACRAWL into a model-based reinforcement learning (RL) algorithm that learns change rates on the fly.

For a source $w$, suppose the tracker observes binary change indicators $\{z_j\}_{j=1}^U$, where $\{t_j\}_{j=0}^U$ are observation times and $z_j = 1$ iff $w$ changed since $t_{j-1}$ *at least* once. Consider two cases:

**Incomplete change observations for $w$.** Here, the tracker generates the sequence $\{z_j\}_{j=1}^U$ for each source $w$ by crawling it. If $z_j = 1$, the tracker still doesn't know exactly how many times the source changed since time $t_{j-1}$. Denoting $a_{t_j} = t_j - t_{j-1}, j \ge 1$, $\hat{\Delta}$ that solves

$$\sum_{j:z_j=1} \frac{a_j}{e^{a_j \Delta} - 1} - \sum_{j:z_j=0} a_j = 0, \qquad (10)$$

is an MLE of $\Delta$ for the given source [11]. The l.h.s. of the equation is monotonically decreasing in $\Delta$, so $\hat{\Delta}$ can be efficiently found numerically. This estimator is consistent under mild conditions [11], e.g., if the sequence $\{a_j\}_{j=1}^\infty$ doesn't converge to 0, i.e., if the observations are spaced apart.

**Complete change observations for** $w$.
In this case, for all $j$, $z_j = 1$: an observation indicating *exactly one* change arrives on every change. Here a consistent MLE of $\Delta$ is the observation rate [30]:

$$\hat{\Delta} = (U+1)/t_U, \qquad (11)$$

LAMBDALEARNANDCRAWL, a model-based RL version of LAMBDACRAWL that uses these estimators to learn model parameters simultaneously with scheduling is presented in Algorithm 4. It operates in epochs of length $T_{epoch}$ time units each (lines 3-13). At the start of each epoch $n$, it calls LAMBDACRAWL (Algorithm 3) on the available $\vec{\hat{\Delta}}_{n-1}$ change rate estimates to produce a policy $(\vec{\rho}_n^*, \vec{p}_n^*)$ optimal with respect to them (line 4). Executing this policy during the current epoch, for the time period of $T_{epoch}$, and recording the observations extends the observation history (lines 7-8). (Note though that for sources $w \in W^o$, the observations don't depend on the policy.) It then re-estimates change rates using a suffix of the augmented observation history (lines 10-13). Under mild assumptions, LAMBDALEARNANDCRAWL converges to the optimal policy:

**Proposition 8.** LAMBDALEARNAND-CRAWL *(Algorithm 4) converges in probability to the optimal policy under the true change rates* $\vec{\Delta}$*, i.e.,*

---

**Algorithm 4:** LAMBDALEARNANDCRAWL: finding optimal crawl scheduling policy $\pi^* \in \Pi^\ominus$ (Problem 3) under initially unknown change model

**Input:** $R > 0$ – bandwidth;
$\vec{\mu} > 0, \vec{\hat{\Delta}}_0 > 0$ – importance and initial change rate guesses
$\epsilon^{\text{no-obs}}, \epsilon > 0$ – desired precisions
$T_{epoch} > 0$ – duration of an epoch
$N_{epochs} > 0$ – number of epochs
$S(n)$ – for each epoch $n$, observation history suffix length for learning $\vec{\Delta}$ in that epoch

1 // $obs\_hist[S(n)]$ is $S(n)$-length observation history suffix
2 $obs\_hist \leftarrow ()$
3 **foreach** $1 \le n \le N_{epochs}$ **do**
4    $\vec{\rho}_n^*, \vec{p}_n^* \leftarrow$ LAMBDACRAWL$(R, \vec{\mu}, \vec{\hat{\Delta}}_{n-1}, \epsilon^{\text{no-obs}}, \epsilon)$
5    // $\vec{Z}_{new}$ holds observations for all sources from start to
6    // end of epoch $n$. Execute policy $(\vec{\rho}_n^*, \vec{p}_n^*)$ to get it
7    $\vec{Z}_{new} \leftarrow ExecuteAndObserve(\vec{\rho}_n^*, \vec{p}_n^*, T_{epoch})$
8    $Append(obs\_hist, \vec{Z}_{new})$
9    // Learn new $\vec{\Delta}$ estimates using Eqs. 10 and 11
10    **foreach** $w \in W^-$ **do**
11      $\hat{\Delta}_{n_w} \leftarrow Solve(\sum_{j:z_{jw}=1} \frac{a_j}{e^{a_j\Delta}-1} + \frac{0.5}{e^{0.5\Delta}-1} - \sum_{j:z_{jw}=0} a_j - 0.5 = 0, obs\_hist[S(n)])$
12    **foreach** $w \in W^o$ **do**
13      $\hat{\Delta}_{n_w} \leftarrow Solve(\frac{U_{S(n)}+0.5}{S(n)+0.5}, obs\_hist[S(n)])$

---

$\lim_{N_{epochs} \to \infty}(\vec{\rho}_{N_{epochs}}, \vec{p}_{N_{epochs}}) = (\vec{\rho}^*, \vec{p}^*)$*, if* $\vec{\Delta}$ *is stationary and* $S(n)$*, the length of the history's training suffix, satisfies* $S(N_{epoch}) = length(obs\_hist)$*.*

*Proof.* See the Supplement. It follows from the consistency and positivity of the change rate estimates, as well as LAMBDACRAWL's optimality ∎

LAMBDALEARNANDCRAWL in practice requires attention to several aspects:

***Stationarity of*** $\vec{\Delta}$***.*** Source change rates may vary with time, so the length of history suffix for estimating $\vec{\Delta}$ should be shorter than the entire available history.

***Singularities of*** $\vec{\hat{\Delta}}$ ***estimators.*** The MLE in Eq. 10 yields $\hat{\Delta}_w = \infty$ if all crawls detect a change (the r.h.s. is 0). Similarly, Eq. 11 produces $\hat{\Delta}_w = 0$ if no observations about $w$ arrive in a given period. To avoid these singularities without affecting consistency, we smooth the estimates by adding imaginary observation intervals of length 0.5 to Eq. 10 and imaginary 0.5 observation to Eq. 11 (lines 11,13).

***Number of parameters.*** Learning a change rate separately for each source can be slow. Instead, we can generalize change rates across sources (e.g., [12]). Alternatively, sometimes we can avoid learning for most pages altogether:

**Proposition 9.** *Suppose the tracker's set of sources* $W^-$ *is such that for some constant* $c > 0$*,* $\frac{\mu_w}{\Delta_w} = c$ *for all* $w \in W^-$*. Then minimizing harmonic penalty under incomplete change observations (Problem 1) has* $\rho_w^* = \frac{\mu_w R^-}{\sum_{w' \in W^-} \mu_{w'}}$*.*

*Proof.* See the Supplement. The proof proceeds by plugging in $\Delta_w = \frac{1}{c}\mu_w$ into Eq. system 5. ∎

Thus, if the importance-to-change-rate ratio is roughly equal across all sources, then their crawl rates don't depend on change rates or even the ratio constant itself. Thus, we don't need to learn them for sources $w \in W^-$ and can hope for faster convergence, although for some quality loss (see Section 7).

## 6 Related work

**Scheduling for Posting, Polling, and Maintenance.** Besides monitoring information sources, mathematically related settings arise in smart broadcasting in social networks [19, 31, 33, 36], personalized teaching [31], database synchronization [14], and job and maintenance service scheduling [1, 3, 15, 16]. In web crawling context (see Olston & Najork [22] for a survey), the closest works are [10], [35], [25], and [2]. Like [10] and [2], we use Lagrange multipliers for optimization, and adopt the Poisson change model of [10] and many works since. Our contributions differ from prior art in several ways: (1) optimization objectives (see below) and guarantees; (2) special crawl scheduling under complete change observations; (3) reinforcement learning of model parameters during crawling.

**Optimization objectives.** Our objective falls in the class of convex separable resource allocation problems [17]. So do most other related objectives: binary freshness/staleness [2, 10], age [8], and embarrassment [35]. The latter is implemented via specially constructed importance scores [35], so our algorithms can be used for it too. Other separable objectives include information longevity [23]. In contrast, Pandey & Olston [25] focus on an objective that depends on user behavior and cannot be separated into contributions from individual sources. While intuitively appealing, their measure can be optimized only via many approximations [25], and the algorithm for it is ultimately heuristic.

**Acquiring model parameters.** Importance can be defined and quickly determined from information readily available to search engines, e.g., page relevance to queries [35], query-independent popularity such as PageRank [24], and other features [25, 28]. Learning change rates is more delicate. Change rate estimators we use are due to [11]; our contribution in this regard is integrating them into crawl scheduling while providing theoretical guarantees, as well as identifying conditions when estimation can be side-stepped using an approximation (Prop. 9). While many works adopted the homogeneous Poisson change process [2, 8, 9, 10, 11, 12, 35], its non-homogeneous variant [14], quasi-deterministic [35], and general marked temporal point process [31] change models were also considered. Change models can also be inferred via generalization using source co-location [12] or similarity [28].

**RL.** Our setting could be viewed as a *restless* multi-armed bandit (MAB) [34], a MAB type that allows an arm to change its reward/cost distribution without being pulled. However, no known restless MAB class allows arms to incur a cost/reward without being pulled, as in our setting. This distinction makes existing MAB analysis such as [18] inapplicable to our model. RL with events and policies obeying general marked temporal point processes was studied in [31]. However, it relies on DNNs and as a result doesn't provide guarantees of convergence, optimality, other policy properties, or a mechanism for imposing strict constraints on bandwidth, and is far more expensive computationally.

## 7 Empirical evaluation

Our experimental evaluation assesses the relative performance of LAMBDACRAWL, LAMBDACRAWLAPPROX, and existing alternatives, evaluates the benefit of using completes change observations, and shows empirical convergence properties of RL for crawl scheduling (Sec. 5). **Please refer to the Supplement, Sec. 9, for details of the experiment setup. All the data and code we used are available at https://github.com/microsoft/Optimal-Freshness-Crawl-Scheduling**.

**Metrics.** We assessed the algorithms in terms of two criteria. One is the *harmonic policy cost* $J_h^\pi$, defined as in Eq. 1 with $C(n)$ as in Eq. 2, which LAMBDACRAWL optimizes directly. The other is the *binary policy cost* $J_b^\pi$, also defined as in Eq. 1 but with $C(n) = \mathbb{1}_{n>0}$. It was used widely in previous works, e.g., [2, 10, 35], and is optimized directly by BinaryLambdaCrawl [2]. LAMBDACRAWL doesn't claim optimality for it, but we can still use it to evaluate LAMBDACRAWL's policy.

**Data and baselines.** The experiments used web page change and importance data collected by crawling $18,532,314$ URLs daily for 14 weeks. We compared LAMBDACRAWL (labeled *LC* in the figures), LAMBDACRAWLAPPROX (*LCA*, *LC* with Prop. 9's approximation), and their RL variants *LLC* (Alg. 4) and *LLCA* to *BinaryLambdaCrawl* (*BLC*) [2], the state-of-the-art optimal algorithm for the binary cost $J_b^\pi$. Since *BLC* may crawl-starve sources and hence get $J_h^\pi = \infty$ (see Fig. 1), we also used our own variant, *BLC$\epsilon$*, with the non-starvation guarantee, and its RL flavor *BLLC$\epsilon$*. Finally, we used *ChangeRateCrawl* (*CC*) [10, 35] and *UniformCrawl* (*UC*) [9, 23] heuristics. In each run of an experiment, the bandwidth $R$ was 20% of the total number of URLs used in that run.

**Results.** We conducted three experiments, whose results support the following claims:

*(1)* LAMBDACRAWL*'s harmonic staleness cost $J_h^\pi$ is a more robust objective than the binary cost $J_b^\pi$ widely studied previously: optimizing the former yields policies that are also near-optimal w.r.t. the latter, while the converse is not true.* In this experiment, whose results are shown in Fig. 1, we

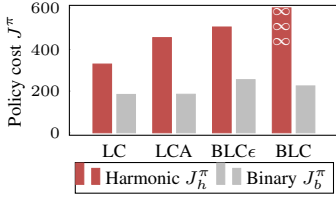
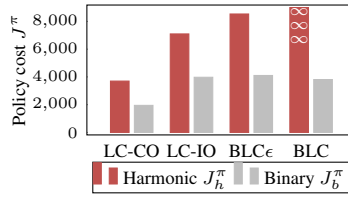
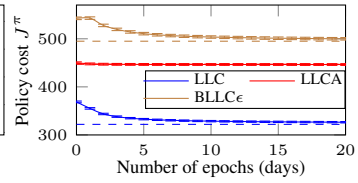

Figure 1: Performance w.r.t. harmonic ($J_h^\pi$) and binary ($J_b^\pi$) policy costs. **Lower bars = better policies.** *LC* is robust to both, but *BLCε* & *BLC* [2] aren't: *LC* ($J_h^\pi$-optimal) beats *BLC* ($J_h^\pi = \infty$) and *BLCε* by 35% w.r.t. $J_h^\pi$, but *BLC* ($J_b^\pi$-optimal for incomplete-change-observation URLs) and *BLCε* don't beat *LC/LCA* w.r.t. $J_b^\pi$. *CC* ($J_h^\pi = 2144, J_b^\pi = 963$) and *UC* ($J_h^\pi = 1268, J_b^\pi = 628$) did poorly and were omitted from the plot.

Figure 2: Benefit of using complete change observations. Here we use only the URLs that provide them (4% of our dataset), via sitemaps and other signals. On this URL subset, LAMBDACRAWL reduces to LC-ComplObs (*LC-CO*, Alg. 2) that heeds these signals, while LC-IncomplObs (*LC-IO*, Alg. 1), *BLC*, and *BLCε* ignore them. As a result, *LC-CO*'s policy cost both w.r.t. $J_h^\pi$ and $J_b^\pi$ is at least 50% lower (!) than the other algorithms'.

Figure 3: Convergence of the RL-based *LLC*, *LLCA*, and *BLLCε* initialized with uniform change rate estimates of 1/day. Dashed lines show asymptotic policy costs ($J_h^\pi$ of *LC*, *LCA*, and *BLCε* from Fig. 1); plots have confidence intervals. *LLC* converges notably faster than *BLLCε*, *LLCA* even more so, as it learns fewer parameters. *LLCA*'s asymptotic policy is worse than *LLC*'s but better than *BLLCε*'s, especially w.r.t. binary cost $J_b^\pi$ (Fig. 1).

assumed known change rates. To obtain them, we applied the change rate estimators in Eqs. 10 and 11 to all of the 14-week crawl data for 18.5M URLs, and used their output as ground truth. Policies were evaluated using equations in Props. 1, 4 to get $J_h^\pi$, and Eqs. 12, 13 in the Supplement to get $J_b^\pi$.

*(2) Utilizing complete change observations as* LAMBDACRAWL *does when they are available makes a very big difference in policy cost.* Per Fig. 1, *LC* outperforms *BLC* even in terms of binary cost $J_b^\pi$, w.r.t. which *BLC* gives an optimatlity guarantee *as long as all URLs have only incomplete change observations*. This raises the question: can *LC*'s and *LCA*'s specialized handling of the complete-observation URLs, a mere 4% of our dataset, explain their overall performance advantage?

The experiment results in Fig. 2 suggest that this is the case. Here we used only the aforementioned URLs with complete change observations. On this URL set, *LC* reduces to *LC-CO* (Alg. 2) and yields a 2× reduction in harmonic cost $J_h^\pi$ compared to treating these URLs conventionally as *LC-IO* (Alg. 1), *BLC*, and *BLCε* do. On the full 18.5M set of URLs, *LC* crawls its complete-observability subset even more effectively by allocating to it a disproportionately large fraction of the overall bandwidth.

Although its handling of complete-observation URLs gives *LC* an edge over alternatives, note that in the hypothetical situation where *LC* treats these URLs conventionally, as reflected in the *LC-IO*'s plot in Fig. 2, it is still at par with *BLC* and *BLCε* w.r.t. $J_b^\pi$, and markedly outperforms them w.r.t. $J_h^\pi$.

*(3) When source change rates are initially unknown, the approximate* LLCA *converges faster w.r.t. $J_h^\pi$ than the optimal* LLC*, but at the cost of higher asymptotic policy cost.* Interestingly, *LLCA*'s approximation (Prop. 9) only weakly affects its asymptotic performance w.r.t. binary cost $J_b^\pi$ (Fig. 1). These factors and algorithm simplicity make this approximation a useful tradeoff in practice.

This experiment, whose analysis is presented in Fig. 3, compared *LLC*, *LLCA*, and *BLLCε* in settings where URL change rates have to be learned on the fly. We chose 20 100,000-URL subsamples of our 18.5M-URL dataset randomly with replacement, and used them to simulate 20 21-day runs of each algorithm starting with change rate estimates of 1 change/day for each URL. We used "ground truth" change rates to generate change times for each URL. Every simulated day (epoch; see Alg. 4), each algorithm re-estimated change rates from observations, which were sampled according to the algorithm's current policy, of simulated URL changes. For the next day, it reoptimized its policy for the new rate estimates, and this policy was evaluated with equations in Props. 1, 4 *under the ground-truth rates*. Each algorithm's policy costs across 20 episodes were averaged for each day.

# 8    Conclusion

We have introduced a new optimization objective and a suite of efficient algorithms for it to address the freshness crawl scheduling problem faced by services from search engines to databases. In particular, we have presented LAMBDALEARNANDCRAWL, which integrates model parameter learning with scheduling optimization. To provide theoretical convergence rate analysis in the future, we intend to frame this problem as a restless multi-armed bandit setting [18, 34].

**Acknowledgements.** We would like to thank Lin Xiao (Microsoft Research) and Junaid Ahmed (Microsoft Bing) for their comments and suggestions regarding this work.

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
