[Supplementary Material]

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

# SUPPLEMENT

## 9 Details of the Experiments and Additional Plots

### 9.1 Dataset, Implementation, Hardware

For the dataset, we crawled $18,532,314$ URLs daily over 14 weeks to estimate their change rates reliably using Equations 10 and 11. Some of the URL crawls on some days failed for reasons ranging from crawler's internal errors to the URL host being temporarily unavailable, so many URLs were crawled fewer than $14 \cdot 7 = 98$ times. At the same time, some URLs were crawled more often as part of the crawler's other workloads.

These URLs are data sources for the knowledge base of a major virtual assistant. The knowledge base uses special information extractors to get important information out of these pages. To determine if a page changed, we ran the same information extractors on it every time we crawled it and considered the page as changed if the extracted information changed.

Figure 4: Ranking of $18,532,314$ URLs in our dataset by their $\frac{\mu_w}{\Delta_w}$ ratio.

Figure 5: Importance score histogram for URLs in our dataset. The distribution has a big skew, with most pages having importance less than 1000.

Figure 6: (Poisson) change rate distribution for URLs in our dataset. Most URLs change once in a few days.

4% of the URLs in the dataset had near-complete change observations that we obtained by frequently crawling reliable sitemaps associated with these URLs and using other signals. The near-completeness is due to the fact that we considered a web page changed only if certain information extracted from it changed, which is usually different from a sitemap maintainer's notion of change. As a result, a sitemap could report a change that we wouldn't consider as one and, conversely, could fail to report changes important to us.

We set the URL importance scores $\mu_w$ to values defined by the production crawler based on PageRank and popularity.

Proposition 9 suggests that the performance gap between LAMBDACRAWL and LAMBDACRAWLAP-PROX depends on the distribution of the ratios $\frac{\mu_w}{\Delta_w}$ across the set of sources $W$: if they are all equal, the policy cost of LAMBDACRAWL and LAMBDACRAWLAPPROX should be the same. As Figure 4 shows, these ratios are similar across much of our dataset, but clearly non-uniform in the head and tail of the ranking. Figures 5 and 6 show the dataset's importance and change rate distributions.

In each run of an algorithm, **the bandwidth constraint was set to 20% of the number of pages used in that run.**

All algorithms used in the experiments were implemented in Python and run on a Windows 10 laptop with 16GB RAM and an Intel quad-core 2.11GHz i7-8650U CPU. The implementations are available at https://github.com/microsoft/Optimal-Freshness-Crawl-Scheduling.

## 9.2 Evaluation metrics

To evaluate the algorithms' performance, we used two metrics:

- **The harmonic policy cost** $J_h^\pi$ as in Equation 1 with $C(n)$ as in Equation 2.
- **The binary policy cost** $J_b^\pi$ as in Equation 1 with $C(n)$ defined as

$$C(n) = \mathbb{1}_{n>0}$$

.

This policy cost objective was studied in several works including [2, 10]. Some prior research considered its finite-horizon [35] and discrete-time versions [2]. Note that some of these papers formulated their objective as *maximizing freshness*, whereby the agent is *rewarded* for each time unit when the number of accumulated changes at a source is 0. Maximizing this objective means minimizing binary staleness (although the two aren't necessarily negations of each other!) Thus, the two are equivalent and we don't distinguish between them in the paper.

Actually using $J_b^\pi$ to evaluate policy requires deriving its parameterization in terms of policy $\pi$'s crawl rates $\vec{\rho}$ and crawl probabilities $\vec{\pi}$, analogously to Propositions 1 and 4 for the harmonic cost $J_h^\pi$. By following the steps in the proofs of these propositions, we derived the following expressions for $J_b^\pi$:

$$J_b^\pi = \sum_{w \in W^-} \frac{\mu_w \Delta_w}{\Delta_w + \rho_w} \quad \text{for pages } w \in W^- \tag{12}$$

$$J_b^\pi = \sum_{w \in W^o} \mu_w (1 - p_w) \quad \text{for pages } w \in W^o \tag{13}$$

Note that Equation 12 is similar to the equation in Azar et al. [2] for evaluating the *freshness reward* of policy $\pi$.

**For each experiment, we report the values of $J_h^\pi$ and $J_b^\pi$ normalized by the number of URLs used in that experiment.**

## 9.3 Algorithms

In the experiments description in Section 7, we refer to the following algorithms used in the empirical evaluation:

LAMBDACRAWL (*LC*), as in Algorithm 3.

LAMBDACRAWL-COMPLOBS (LC-ComplObs, *LC-CO*), as in Algorithm 2.

LAMBDACRAWL-INCOMLOBS (LC-IncomplObs, *LC-IO*), as in Algorithm 1.

*BinaryLambdaCrawl* (*BLC*) is the name we give to the state-of-the-art, optimal algorithm proposed by Azar et al. [2] for minimizing binary staleness $J_b^\pi$. *BLC* is one of two major baselines for LAMBDACRAWL in our experiments.

The difference between *BLC*'s objective $J_b^\pi$ and *LC*'s objective $J_h^\pi$ is crucial in practice, because minimizing binary staleness $J_b^\pi$ generally yields $\vec{\rho}^*$ with $\rho_w^* = 0$ for many sources even if they have $\Delta_w > 0$. This effectively tells the tracker to ignore changes to these sources — an unacceptable strategy in real applications. Indeed, harmonic penalty (Equation 1) assigns $J_h^\pi = \infty$ to such strategies.

*BinaryLambdaCrawl*($\epsilon$) (*BLC$\epsilon$*) Vanilla *BLC*'s lack of non-starvation guarantees makes comparing it to LAMBDACRAWL in terms of $J_h^\pi$ uninsightful, because for *BLC*, $J_h^\pi$ is usually $\infty$. To address this

issue, and simultaneously make *BLC* more practical, we modified *BLC* to enforce the non-starvation guarantee. The resulting algorithm is *BLCε*.

Namely, *BLCε* accepts a parameter $\epsilon \in [0, 1]$. Initially, it operates exactly like *BLC* to find $\vec{\rho^*}$ optimal w.r.t. the binary cost $J_b^\pi$. Then it finds all sources $w$ for which

$$\rho_w^* < \epsilon R/|W|,$$

sets $\rho_w^* = \epsilon R/|W|$ for each of them, and re-solves the problem over the remaining sources and bandwidth again using *BLC*. Thus, *BLCε* uniformly distributes a small fraction of the bandwidth to sources that would otherwise get no or little crawl rate allocated to it.

The original, optimal *BLC* can be viewed as *BLC(0)*. Any $\epsilon > 0$ results in suboptimality w.r.t. $J_b^\pi$ while ensuring that $J_h^\pi < \infty$. For the experiments, we did a parameter sweep to determine $\epsilon$ that resulted in *BLCε*'s best performance w.r.t. LAMBDACRAWL's objective $J_h^\pi$. **The best value we found for our dataset, and used in all the experiments, is $\epsilon = 0.4$.**

*UniformCrawl* (*UC*) [9, 23] is a heuristic that assigns an equal crawl rate to all sources:

$$\rho_w = R/|W|.$$

In spite of its simplicity, in our experiments it outperforms *ChangeRateCrawl*, as predicted by Cho & Garcia-Molina [10] (see the caption of Figure 1).

*ChangeRateCrawl* (*CC*) is the name we give to another heuristic proposed by Cho & Garcia-Molina [10] that sets

$$\rho_w = \frac{\Delta_w R}{\sum_{w' \in W} \Delta_{w'}},$$

thereby crawling sources at a rate proportional to their change rate. Cho & Garcia-Molina [10] pointed out that *ChangeRateCrawl* can be very suboptimal if the set $W$ includes sources that change frequently. This causes *ChangeRateCrawl* to over-commit crawl bandwidth to sources whose changes are near-impossible to keep up with, at the expense of almost ignoring the rest. Our experimental results agree with this observation — *ChangeRateCrawl* turned out to be the weakest-performing algorithm in our experiments.

LAMBDALEARNANDCRAWL (*LLC*), as in Algorithm 4.

LAMBDALEARNANDCRAWLAPPROX (*LLCA*), as in Algorithm 4 with calls to LAMBDACRAWL-INCOMLOBS in LAMBDACRAWL replaced by setting

$$\rho_w = \frac{\mu_w R}{\sum_{w' \in W^-} \mu_{w'}}.$$

per Proposition 9.

*BinaryLambdaLearnAndCrawl(ε)* (*BLLCε*), the reinforcement learning version of *BLCε* where *BLCε* replaces LAMBDACRAWL in Algorithm 4. This is also the natural RL adaptation of pure *BLC* [2], which otherwise would need a dedicated exploration parameter to ensure data gathering for sources that would get $\rho_w = 0$ under *BLC*'s (currently) optimal policy. As for *BLCε*, we used $\epsilon = 0.4$.

## 9.4 Experiment 1 (Figure 1)

The goal of this experiment was to assess the harmonic objective $J_h^\pi$ that we proposed and the binary objective $J_b^\pi$ widely studied in previous works in terms of robustness: how well do policies optimal w.r.t. one of them behave w.r.t. the other, and vice versa?

In this experiment, we assumed known change rates. To obtain them, we inferred them with estimators in Equations 10 and 11 from our entire 14-week crawl data for 18.5M URLs, and used the resulting estimates as ground truth. Policies were evaluated by plugging in these change rates and policy parameters into the equations in Propositions 1 and 4 and into Equations 12 and 13.

As Figure 1 shows, the harmonic penalty $J_h^\pi$ we propose is a more flexible choice of objective than the binary penalty $J_b^\pi$. LAMBDACRAWL, optimal w.r.t. it, significantly outperforms $BLC\epsilon$ and $BLC$ w.r.t. it, and even the approximate LAMBDACRAWLAPPROX performs at par with $BLC\epsilon$ according to this objective. What is even more surprising, LAMBDACRAWL also manages to outperform $BLC$ on the binary objective $J_b^\pi$, for which $BLC$ is optimal *if all URLs have only incomplete change observations*. That is, no matter which objective we trust, optimizing for $J_h^\pi$ yields excellent results.

As a side note, the *UniformCrawl* and *ChangeRateCrawl* heuristics were outperformed by a large margin by the above methods w.r.t. both objectives.

## 9.5 Experiment 2 (Figure 2)

One of our contributions is a mechanism for taking advantage of complete remote change observations (Algorithm 2). $BLC\epsilon$, $BLC$, *UniformCrawl*, and *ChangeRateCrawl* don't have it, and treat all pages as if the only change observations for them came from crawling. While only 4% of web pages in our dataset have an (approximately) complete observation history, can LAMBDACRAWL's and LAMBDACRAWLAPPROX's advantage on them explain the performance gap in experiment 1?

Figure 2 indicates that these URLs are indeed responsible for a significant fraction of LAMBDACRAWL's advantage. In this experiment, we focused only on URLs with complete change observations. Like in the previous experiment, we assumed perfect model knowledge using the previously obtained change rate estimates and estimated policy performance using Propositions 1, 4 and Equations 12 and 13. Treating these URLs as complete-observation URLs, as LAMBDACRAWL-COMPLOBS does, resulted in nearly 5-fold reduction in policy cost, compared to treating these URLs under the conventional change observation model.

Although this gives LAMBDACRAWL an edge over previously proposed techniques, note that even when LAMBDACRAWL treats these URLs conventionally (denoted by the LAMBDACRAWL-INCOMLOBS (*LC-IO*) plot in Figure 2), it still noticeably outperforms $BLC$ and $BLC\epsilon$ w.r.t. $J_h^\pi$ while holding its own against them w.r.t. $J_b^\pi$.

## 9.6 Experiment 3 (Figure 3)

Last but not least, we analyze the reinforcement learning variants of LAMBDACRAWL, LAMBDACRAWLAPPROX, and *BLLC$\epsilon$*. Our motivation for the approximation in Proposition 9 was reducing the number of parameters LAMBDALEARNANDCRAWL has to learn in order to speed up convergence. In this experiment, we explore the tradeoff between the resulting gain in learning speed and the concomitant loss in solution quality.

The evaluation was done in a series of 20 simulated episodes for each algorithm, each episode being executed on a randomly chosen 100,000-URL subsample of the 18.5M URLs (each subsample was used once be each of the three algorithms.) In each episode, we simulated a 21-day run of *LLC*, *LLCA*, and *BLLC$\epsilon$* starting with change rate estimates of 1 change per day for each URL. One epoch (see Algorithm 4) corresponded to 1 day. That is, every (simulated) day each of these algorithms re-estimated the change parameters using the simulated observation data (the simulated data wasn't shared among the algorithms). The simulated data was generated by sampling page changes using the ground truth change rates obtained in previous experiments and sampling page crawls from each algorithm's current policy. At the end of each day, each algorithm reoptimized its policy for the new estimates, and this policy was evaluated using the aforementioned equations *on the ground truth change rates*. For each algorithm, averaged the policy costs for each day across all 20 episodes.

Figure 3 demonstrates that LAMBDALEARNANDCRAWLAPPROX indeed converges quicker than the other algorithms, with its asymptotic performance being better than *BLLC$\epsilon$*'s but falling short

of LAMBDALEARNANDCRAWL's. *BLLC$\epsilon$*'s convergence was the slowest. It could potentially be improved by choosing a larger $\epsilon$ at the beginning and gradually "cooling" it. LAMBDALEARNAND-CRAWL's advantage, besides convergence speed and asymptotic performance, is that it converges quickly without such parameter tuning. LAMBDALEARNANDCRAWLAPPROX, however, can also be a viable alternative to it, given its simplicity, fast convergence, and good (though suboptimal) policy quality.

## 10 Proofs

**PROOF OF PROPOSITION 1.**

First we rearrange Equation 1 as follows, dropping the distributions under the expectation and using $W$ instead of $W^-$ throughout the proof to make the notation less cumbersome:

$$J^\pi = \lim_{T\to\infty} \mathop{\mathbb{E}}_{\substack{CrSeq\sim\pi, \\ ChSeq\sim P(\vec{\Delta})}} \left[ \frac{1}{T}\int_0^T \left( \sum_{w\in W} \mu_w C(N_w(t)) \right) dt \right]$$

$$= \lim_{T\to\infty} \frac{1}{T}\int_0^T \left( \sum_{w\in W} \mu_w \mathbb{E}\left[C(N_w(t))\right] \right) dt$$

Then we use the definition of expectation, chain rule of probabilities, and variable $t_{prev}$ to denote the time when source $w$ was last crawled before time $t$ (although $t_{prev}$ is specific to each source $w$, for clarity of notation we make this implicit):

$$J^\pi = \lim_{T\to\infty} \frac{1}{T}\int_0^T \left( \sum_{w\in W} \mu_w \mathbb{E}[C(N_w(t))] \right) dt$$

$$= \lim_{T\to\infty} \frac{1}{T}\int_0^T \left( \sum_{w\in W} \mu_w \sum_{m=0}^\infty \left( C(m)\cdot\mathbb{P}[N_w(t)=m] \right) \right) dt$$

$$= \lim_{T\to\infty} \frac{1}{T}\int_0^T \left( \sum_{w\in W} \mu_w \sum_{m=0}^\infty \left( C(m)\int_0^t \mathbb{P}[N_w(t)=m \mid t-t_{prev}=T']\,\mathbb{P}[t-t_{prev}=T']dT' \right) \right) dt$$

Now using the fact that our policy is a set of Poisson processes with parameters $\rho_w$ and page changes are governed by another set of Poisson processes with parameters $\Delta_w$, we can plug in appropriate expressions for the probabilities:

$$J^\pi = \lim_{T\to\infty} \frac{1}{T}\int_0^T \left( \sum_{w\in W} \mu_w \sum_{m=0}^\infty \left( C(m)\int_0^t \left( \frac{(T'\Delta_w)^m e^{-T'\Delta_w}}{m!} \right)\left( \rho e^{-\rho_w T'} \right)dT' \right) \right) dt$$

$$= \lim_{T\to\infty} \frac{1}{T}\int_0^T \left( \sum_{w\in W} \mu_w \sum_{m=0}^\infty \left( \frac{C(m)\rho_w\Delta_w^m}{m!}\int_0^t T'^m e^{-(\rho_w+\Delta_w)T'}dT' \right) \right) dt$$

We do a variable substitution $u=(\rho_w+\Delta_w)T'$, so $T'=\frac{u}{\Delta_w+\rho_w}$ and $dT'=\frac{du}{\Delta_w+\rho_w}$:

$$J^\pi = \lim_{T\to\infty} \frac{1}{T} \int_0^T \left( \sum_{w\in W} \mu_w \sum_{m=0}^{\infty} \left( \frac{C(m)\rho_w \Delta_w^m}{m!} \int_0^t \left(\frac{u}{\Delta_w + \rho_w}\right)^m e^{-u} \frac{du}{\Delta_w + \rho_w}\right)\right) dt$$

$$= \lim_{T\to\infty} \frac{1}{T} \int_0^T \left( \sum_{w\in W} \mu_w \sum_{m=0}^{\infty} \left( \frac{C(m)}{m!} \left(\frac{\rho_w}{\Delta_w + \rho_w}\right) \left(\frac{\Delta_w}{\Delta_w + \rho_w}\right)^m \int_0^t u^m e^{-u} du\right)\right) dt$$

Consider $F(m,t) = \int_0^t u^m e^{-u} du$. By definition of gamma functions, $F(m,t) = \Gamma(m+1) - \Gamma(m+1,t) = m! - \Gamma(m+1,t)$. Recalling that $C(m) = H(m)$ for $m > 0$ and $C(0) = 0$ (Equation 2), we get

$$J^\pi = \lim_{T\to\infty} \frac{1}{T} \int_0^T \left( \sum_{w\in W} \mu_w \sum_{m=1}^{\infty} \left( \frac{H(m)}{m!} \left(\frac{\rho_w}{\Delta_w + \rho_w}\right) \left(\frac{\Delta_w}{\Delta_w + \rho_w}\right)^m (m! - \Gamma(m+1,t))\right)\right) dt$$

$$= \lim_{T\to\infty} \frac{1}{T} \left( \sum_{w\in W} \left(\frac{\mu_w \rho_w}{\Delta_w + \rho_w}\right) \left( \int_0^T \sum_{m=1}^{\infty} \left( H(m) \left(\frac{\Delta_w}{\Delta_w + \rho_w}\right)^m\right) dt - \right.\right.$$

$$\left.\left. - \int_0^T \sum_{m=1}^{\infty} \left( \frac{H(m)}{m!} \left(\frac{\Delta_w}{\Delta_w + \rho_w}\right)^m \Gamma(m+1,t)\right) dt\right)\right) \qquad (14)$$

Now consider for each $w \in W$ functions $G(T) = \int_0^T \sum_{m=1}^{\infty} \left( H(m) \left(\frac{\Delta_w}{\Delta_w+\rho_w}\right)^m\right) dt$, $R(t) = \int_0^T \sum_{m=1}^{\infty} \left( \frac{H(m)}{m!} \left(\frac{\Delta_w}{\Delta_w+\rho_w}\right)^m \Gamma(m+1,t)\right) dt$. Consider $\lim_{T\to\infty} \frac{1}{T}(G(T) - R(T))$ so that

$$J^\pi = \sum_{w\in W} \left(\mu_w \left(\frac{\rho_w}{\Delta_w + \rho_w}\right) \left(\lim_{T\to\infty} \frac{1}{T}(G(T) - R(T))\right)\right) \qquad (15)$$

Thus, if $\lim_{T\to\infty} \frac{1}{T}(G(T) - R(T))$ exists, is finite, and we can compute it, we can compute $J^\pi$ as well. Focusing on $G(T)$ and recalling that $\sum_{m=1}^{\infty} H(m)x^m = -\frac{\ln(1-x)}{1-x}$ for $|x| < 1$, we see that $G(T) = -\frac{\ln(\frac{\rho_w}{\Delta_w+\rho_w})}{\frac{\rho_w}{\Delta_w+\rho_w}}T$, so $\lim_{T\to\infty} \frac{G(T)}{T} = -\frac{\ln(\frac{\rho_w}{\Delta_w+\rho_w})}{\frac{\rho_w}{\Delta_w+\rho_w}}$ exists. Focusing on $R(T)$, we see that since $m! < \Gamma(m+1,t)$ for any $t > 0$ and since $R(T)$ is an integral of a non-negative function, we have $0 \le R(T) \le G(T) < \infty$, so $\lim_{T\to\infty} \frac{R(T)}{T}$ exists as well. Therefore, we can write $\lim_{T\to\infty} \frac{1}{T}(G(T) - R(T)) = \lim_{T\to\infty} \frac{G(T)}{T} - \lim_{T\to\infty} \frac{R(T)}{T}$.

We already know $\lim_{T\to\infty} \frac{G(T)}{T}$. To evaluate $\lim_{T\to\infty} \frac{R(T)}{T}$, we upper-bound $R(T)$. Again using the fact that it is an integral of a non-negative function, we have

$$R(T) = \int_0^T \sum_{m=1}^{\infty} \left( \frac{H(m)}{m!} \left(\frac{\Delta_w}{\Delta_w + \rho_w}\right)^m \Gamma(m+1,t)\right) dt$$

$$< \int_0^{\infty} \sum_{m=1}^{\infty} \left( \frac{H(m)}{m!} \left(\frac{\Delta_w}{\Delta_w + \rho_w}\right)^m \Gamma(m+1,t)\right) dt$$

$$= \sum_{m=1}^{\infty} \left( \frac{H(m)}{m!} \left(\frac{\Delta_w}{\Delta_w + \rho_w}\right)^m \int_0^{\infty} \Gamma(m+1,t) dt\right)$$

$$= \sum_{m=1}^{\infty} \left( \frac{H(m)}{m!} \left(\frac{\Delta_w}{\Delta_w + \rho_w}\right)^m \Gamma(m+2)\right)$$

$$= \sum_{m=1}^{\infty} \left( (m+1)H(m) \left(\frac{\Delta_w}{\Delta_w + \rho_w}\right)^m\right)$$

We have used the fact that $\int_0^\infty \Gamma(m+1,t)dt = \Gamma(m+2) = (m+1)!$. The series $\sum_{m=1}^\infty (m+1)H(m)x^m$ converges for $|x| < 1$ to some limit $L > 0$, so we have $\lim_{T\to\infty} \frac{R(T)}{T} \leq \lim_{T\to\infty} \frac{L}{T} = 0$ and $\lim_{T\to\infty} \frac{1}{T}(G(T) - R(T)) = -\frac{\ln(\frac{\rho_w}{\Delta_w + \rho_w})}{\frac{\rho_w}{\Delta_w + \rho_w}}$. Plugging this back into Equation 15, we get

$$J^\pi = -\sum_{w\in W} \mu_w \ln\left(\frac{\rho_w}{\Delta_w + \rho_w}\right)$$

∎

**Corollary 1.** *Under the harmonic penalty $C(n)$ (Equation 2), any policy that crawls each source at a fixed Poisson rate $\rho_w$ and assigns $\rho_w > 0$ to each $w$ with $\mu_w, \Delta_w > 0$ is strictly preferable to any such policy that assigns $\rho_w = 0$ to any such source.*

*Proof.* Proposition 1 implies that any policy $\pi$ with $\rho_w = 0$ for any source $w$ with $\mu_w, \Delta_w > 0$ has $J^\pi = \infty$, whereas any $\pi$ with $\rho_w > 0$ for every source $w$ with $\mu_w, \Delta_w > 0$ has $J^\pi < \infty$ ∎

**PROOF OF PROPOSITION 2.** The high-level idea is to apply the method of Lagrange multipliers to Problem 1's relaxation *without* inequality constraints $\vec{\rho} \geq 0$ and show that (a) only one local maximum of this relaxation is within the region given by $\vec{\rho} \geq 0$ – the one satisfying Equation system 5 – and (b) solutions that touch the boundary of this region, i.e., have $\rho_w = 0$ for any $w \in W^-$, are suboptimal. In fact, part (b) follows immediately from Corollary 1, so we only need to solve the relaxation and show part (a).

To apply the method of Lagrange multipliers to the relaxation, we set $f(\vec{\rho}) = \overline{J}^\pi = \sum_{w\in W^-} \mu_w \ln\left(\frac{\rho_w}{\Delta_w + \rho_w}\right)$ and $g(\vec{\rho}) = \sum_{w\in W^-} \rho_w - R$. We need to solve

$$\begin{cases} \nabla f(\vec{\rho}) = \lambda \nabla g(\vec{\rho}) \\ g(\vec{\rho}) = 0. \end{cases}$$

For any $w \in W^-$, we have $\frac{\partial g}{\partial \rho_w} = 1$ and $\frac{\partial f}{\partial \rho_w} = \mu_w \frac{\Delta_w + \rho_w}{\rho_w} \frac{\Delta_w}{(\Delta_w + \rho_w)^2} = \frac{\mu_w \Delta_w}{\Delta_w \rho_w + \rho_w^2}$, so the above system of equations turns into

$$\begin{cases} \frac{\mu_w \Delta_w}{\Delta_w \rho_w + \rho_w^2} = \lambda, & \text{for all } w \in W^- \\ \sum_{w\in W^-} \rho_w = R \end{cases}$$

and therefore

$$\begin{cases} \lambda \rho_w^2 + \lambda \Delta_w \rho_w - \mu_w \Delta_w = 0, & \text{for all } w \in W^- \\ \sum_{w\in W^-} \rho_w = R \end{cases}$$

Solving each quadratic equation separately, we get

$$\begin{cases} \rho_w = \frac{-\Delta_w \pm \sqrt{\Delta_w^2 + \frac{4\mu_w \Delta_w}{\lambda}}}{2}, & \text{for all } w \in W^- \\ \sum_{w\in W^-} \rho_w = R \end{cases}$$

This gives all potential solutions to the relaxation of Problem 1. Now consider the inequality constraints $\vec{\rho} \geq 0$ omitted so far. Observe that any real solution to the above system that has $\rho_w = \frac{-\Delta_w - \sqrt{\Delta_w^2 + \frac{4\mu_w \Delta_w}{\lambda}}}{2}$ for any $w \in W^-$ implies $\rho_w < 0$ for $\mu_w, \Delta_w > 0$, which violates these constraints. Therefore, any solution to Problem 1 itself must satisfy

$$\begin{cases} \rho_w = \frac{-\Delta_w + \sqrt{\Delta_w^2 + \frac{4\mu_w \Delta_w}{\lambda}}}{2}, & \text{for all } w \in W^- \\ \sum_{w \in W^-} \rho_w = R \end{cases}$$

and have $\lambda \geq 0$ (otherwise $\vec{\rho} < 0$, again violating the inequality constraints). Although the first group of equations are non-linear, note that each $\rho_w(\lambda)$ is strictly monotone decreasing in $\lambda$ for $\lambda \geq 0$, so $\sum_{w \in W^-} \rho_w$ is strictly monotone decreasing too implying that $\sum_{w \in W^-} \rho_w(\lambda) = R$ has a unique solution in $\lambda$, and therefore there is a unique $\vec{\rho}$ satisfying the above system of equations. Thus, this $\vec{\rho}$ is the solution to Problem 1 and therefore corresponds to $\pi^* \in \Pi^-$. ∎

## PROOF OF PROPOSITION 3

We start by combining Equation system 5,

$$\begin{cases} \rho_w = \frac{-\Delta_w + \sqrt{\Delta_w^2 + \frac{4\mu_w \Delta_w}{\lambda}}}{2}, & \text{for all } w \in W^- \\ \sum_{w \in W^-} \rho_w = R, \end{cases} \tag{16}$$

into one equation:

$$\sum_{w \in W^-} \frac{-\Delta_w + \sqrt{\Delta_w^2 + \frac{4\mu_w \Delta_w}{\lambda}}}{2} = R$$

Bisection search is guaranteed to converge to *some* solution $\lambda$ of this equation as long as we initialize the search with $\lambda_{lower}, \lambda_{upper}$ s.t. $\lambda \in [\lambda_{lower}, \lambda_{upper}]$. However, all $\lambda_- \leq 0$ that solve Equation system 5 correspond to solutions that have $\vec{\rho} < 0$. At the same time, from the proof of Proposition 2 we know that there is exactly one $\lambda_+ > 0$ that solves Equation system 5, and it corresponds to the (unique) the optimal solution $\vec{\rho}^*$ of Problem 1. We want bisection search to find only this $\lambda_+ > 0$, so we want $\lambda_{lower}, \lambda_{upper}$ s.t. $\lambda_+ \in [\lambda_{lower}, \lambda_{upper}]$ *and* $\lambda_{lower}, \lambda_{upper} > 0$.

To find these bounds, we observe that the l.h.s. of the above equation is monotonically decreasing in $\lambda$ for $\lambda > 0$, so if we find any $\lambda_l > 0$ that guarantees

$$\sqrt{\Delta_w^2 + \frac{4\mu_w \Delta_w}{\lambda_l}} \geq \Delta_w + \frac{2R}{|W|} \text{ for all } w \in W^-,$$

then we have

$$\sum_{w \in W^-} \frac{-\Delta_w + \sqrt{\Delta_w^2 + \frac{4\mu_w \Delta_w}{\lambda_l}}}{2} \geq R$$

and hence $\lambda_l \leq \lambda_+$. To find such $\lambda_l$, we perform a series of algebraic manipulations:

$$\sqrt{\Delta_w^2 + \frac{4\mu_w \Delta_w}{\lambda_l}} \geq \Delta_w + \frac{2R}{|W|} \text{ for all } w \in W^-, \lambda_l > 0$$

$$\Longleftrightarrow \Delta_w^2 + \frac{4\mu_w \Delta_w}{\lambda_l} \geq \left(\Delta_w + \frac{2R}{|W^-|}\right)^2 \text{ for all } w \in W^-, \lambda_l > 0$$

$$\Longleftrightarrow \frac{4\mu_w \Delta_w}{\lambda_l} \geq \frac{4\Delta_w R}{|W^-|} + 4\left(\frac{R}{|W^-|}\right)^2 \text{ for all } w \in W^-, \lambda_l > 0$$

$$\Longleftrightarrow \lambda_l \leq \frac{|W^-|^2 \mu_w \Delta_w}{|W^-|\Delta_w R + R^2} \text{ for all } w \in W^-, \lambda_l > 0$$

$$\Longleftarrow \lambda_l \leq \frac{|W^-|^2 \min_{w\in W^-}\{\mu_w\} \min_{w\in W^-}\{\Delta_w\}}{|W^-|\max_{w\in W^-}\{\Delta_w\} R + R^2}, \lambda_l > 0$$

Since the r.h.s. of this inequality is always positive, we set $\lambda_{lower} = \frac{|W^-|^2 \min_{w\in W^-}\{\mu_w\} \min_{w\in W^-}\{\Delta_w\}}{|W^-|\max_{w\in W^-}\{\Delta_w\} R + R^2}$. An analogous chain of reasoning shows that we can choose $\lambda_{upper} = \frac{|W^-|^2 \max_{w\in W^-}\{\Delta_w\} \max_{w\in W^-}\{\mu_w\}}{|W^-|\min_{w\in W^-}\{\Delta_w\} R + R^2}$.

To establish a bound on the running time, we observe that each iteration of LAMBDACRAWL-INCOMLOBS involves evaluating $\sum_{w\in W^-} \rho_w$, which takes $O(|W^-|)$ time, so by the properties of bisection search the total running time is $O(\log_2(\frac{\lambda_{upper}-\lambda_{lower}}{\epsilon})|W^-|)$.  ∎

**PROOF OF PROPOSITION 4** Let $Ch_w(t)$ denote the total number of changes that have happened at source $w$ in time interval $[0, t]$. As in the proof of Proposition 1, we start with rearranging the cost function in Equation 1, dropping the distributions under the expectation and using $W$ instead of $W^o$ throughout the proof to make the notation less cumbersome. Using the definition of expectation and chain rule of probabilities to get:

$$J^\pi = \lim_{T\to\infty} \mathop{\mathbb{E}}_{\substack{CrSeq\sim\pi, \\ ChSeq\sim P(\vec{\Delta})}} \left[\frac{1}{T}\int_0^T \left(\sum_{w\in W} \mu_w C(N_w(t))\right) dt\right]$$

$$= \lim_{T\to\infty} \frac{1}{T}\int_0^T \left(\sum_{w\in W} \mu_w \mathbb{E}\left[C(N_w(t))\right]\right) dt$$

$$= \lim_{T\to\infty} \frac{1}{T}\int_0^T \left(\sum_{w\in W} \mu_w \sum_{m=0}^\infty \left(C(m) \cdot \mathbb{P}[N_w(t) = m]\right)\right) dt$$

$$= \lim_{T\to\infty} \frac{1}{T}\int_0^T \left(\sum_{w\in W} \mu_w \sum_{c=0}^\infty \sum_{m=0}^\infty \left(C(m) \cdot \mathbb{P}[N_w(t) = m | Ch_w(t) = c]\mathbb{P}[Ch_w(t) = c]\right)\right) dt$$

Since changes at every source $w$ are governed by a Poisson process with rate $\Delta_w$, $\mathbb{P}[Ch_w(t) = c] = \frac{e^{-\Delta_w t}(\Delta_w t)^c}{c!}$. Now, consider $\mathbb{P}[N_w(t) = m | Ch_w(t) = c]$. Recall that whenever source $w \in W^o$ changes, we find out about the change immediately; with probability $p_w$ the policy $\pi \in \Pi^o$ then crawls source $w$ straight away, and with probability $(1 - p_w)$ it waits to make this decision until we find out about $w$'s next change. Therefore, the only way we can have $N_w(t) = m$ is if our policy crawled source $w$ $m + 1$ changes ago *and has not* chosen to crawl $w$ after any of the $m$ changes that happened since, or it hasn't chosen to crawl $w$ since "the beginning of time". Thus, $N_w(t)$ is geometrically distributed, assuming that at least $m + 1$ changes actually happened at source $w$ in the time interval $[0, 1]$. Thus, we have

$$\mathbb{P}[N_w(t) = m | Ch_w(t) = c] = \begin{cases} p_w(1 - p_w)^m, & \text{if } c \geq m + 1 \\ (1 - p_w)^m, & \text{if } c = m \\ 0 & \text{otherwise} \end{cases}$$

Recalling that $C(m) = H(m)$ for $m > 0$ and $C(0) = 0$ (Equation 2) and putting everything together, we have

$$J^\pi = \lim_{T \to \infty} \frac{1}{T} \int_0^T \left( \sum_{w \in W} \mu_w \sum_{c=0}^\infty \sum_{m=0}^\infty \left( C(m) \cdot \mathbb{P}[N_w(t) = m | Ch_w(t) = c] \mathbb{P}[Ch_w(t) = c] \right) \right) dt$$

$$= \lim_{T \to \infty} \frac{1}{T} \int_0^T \left( \sum_{w \in W} \mu_w \sum_{c=1}^\infty \left( \frac{e^{-\Delta_w t}(\Delta_w t)^c}{c!} \right) \left( p_w \sum_{m=1}^{c-1} H(m)(1 - p_w)^m + H(c)(1 - p_w)^c \right) \right) dt$$

$$= \lim_{T \to \infty} \frac{1}{T} \Bigg( \int_0^T \sum_{w \in W} \mu_w \sum_{c=1}^\infty \left( \frac{e^{-\Delta_w t}(\Delta_w t)^c}{c!} \right) \left( p_w \sum_{m=1}^\infty H(m)(1 - p_w)^m \right) dt$$

$$- \int_0^T \sum_{w \in W} \mu_w \sum_{c=1}^\infty \left( \frac{e^{-\Delta_w t}(\Delta_w t)^c}{c!} \right) \left( p_w \sum_{m=c}^\infty H(m)(1 - p_w)^m \right) dt$$

$$+ \int_0^T \sum_{w \in W} \mu_w \sum_{c=1}^\infty \left( \frac{e^{-\Delta_w t}(\Delta_w t)^c}{c!} \right) \left( H(c)(1 - p_w)^c \right) dt \Bigg)$$

Now consider

$$G(T) = \int_0^T \sum_{w \in W} \mu_w \sum_{c=0}^\infty \left( \frac{e^{-\Delta_w t}(\Delta_w t)^c}{c!} \right) \left( p_w \sum_{m=0}^\infty C(m)(1 - p_w)^m \right) dt$$

$$R(T) = \int_0^T \sum_{w \in W} \mu_w \sum_{c=0}^\infty \left( \frac{e^{-\Delta_w t}(\Delta_w t)^c}{c!} \right) \left( p_w \sum_{m=c}^\infty C(m)(1 - p_w)^m \right) dt$$

$$F(T) = \int_0^T \sum_{w \in W} \mu_w \sum_{c=0}^\infty \left( \frac{e^{-\Delta_w t}(\Delta_w t)^c}{c!} \right) \left( C(c)(1 - p_w)^c \right) dt$$

Since

$$J^\pi = \lim_{T \to \infty} \frac{1}{T}(G(T) - R(T) + F(T)),$$

if we show that each of $\lim_{T \to \infty} \frac{G(T)}{T}$, $\lim_{T \to \infty} \frac{R(T)}{T}$, and $\lim_{T \to \infty} \frac{F(T)}{T}$ exists and manage to compute them, then we will know $J^\pi$. The rest of the proof focuses on computing these limits.

**Consider** $G(T)$. Note that $p_w \sum_{m=0}^\infty C(m)(1-p_w)^m = p_w \sum_{m=1}^\infty H(m)(1-p_w)^m$ doesn't depend on $c$. We can use the identity $\sum_{m=1}^\infty H(m)x^m = -\frac{\ln(1-x)}{1-x}$ for $|x| < 1$ to get $p_w \sum_{m=0}^\infty C(m)(1 - p_w)^m = -\ln(p_w)$, so $G(T) = -\int_0^T \left( \sum_{w \in W} \mu_w \sum_{c=0}^\infty \left( \frac{e^{-\Delta_w t}(\Delta_w t)^c}{c!} \right) \ln(p_w) \right) dt$. To simplify $G(T)$ further, note that $\sum_{c=0}^\infty \left( \frac{e^{-\Delta_w t}(\Delta_w t)^c}{c!} \right)$ is just the probability of *any* number of changes occurring in time interval $[0, t]$ under a Poisson process, and therefore equals 1. Thus, $G(T) = -\int_0^T \left( \sum_{w \in W} \mu_w \ln(p_w) \right) dt = -T \sum_{w \in W} \mu_w \ln(p_w)$, so

$$\lim_{T \to \infty} \frac{G(T)}{T} = - \sum_{w \in W} \mu_w \ln(p_w).$$

**Consider** $R(T)$. Observe that $C(m) = H(m) < m$ for $m > 1$, so $p_w \sum_{m=c}^\infty C(m)(1 - p_w)^m < p_w \sum_{m=c}^\infty m(1 - p_w)^m = \frac{(1-p_w)^c(cp_w - p_w + 1)}{p_w}$. Because $(1 - p_w)^c$ decreases in $c$ much faster than $\frac{1}{c(cp_w - p_w + 1)}$ for any fixed $0 < p_w \leq 1$, for any $w \in W$ there is a $c_w^*$ s.t. $p_w \sum_{m=c}^\infty C(m)(1 -$

$p_w)^m < \frac{(1-p_w)^c(cp_w-p_w+1)}{p_w} < \frac{1}{c}$ for any $c > c_w^*$. Let $c^* = \max_{w \in W} c_w^*$. We can then upper-bound $R(T)$ as follows:

$$R(T) = \int_0^T \left( \sum_{w \in W} \mu_w \sum_{c=0}^{\infty} \left( \frac{e^{-\Delta_w t}(\Delta_w t)^c}{c!} \right) \left( p_w \sum_{m=c}^{\infty} C(m)(1-p_w)^m \right) \right) dt = R_1(T) + R_2(T),$$

where

$$R_1(T) = \int_0^T \left( \sum_{w \in W} \mu_w \sum_{c=0}^{c^*-1} \left( \frac{e^{-\Delta_w t}(\Delta_w t)^c}{c!} \right) \left( p_w \sum_{m=c}^{\infty} C(m)(1-p_w)^m \right) \right) dt$$

$$R_2(T) = \int_0^T \left( \sum_{w \in W} \mu_w \sum_{c=c^*}^{\infty} \left( \frac{e^{-\Delta_w t}(\Delta_w t)^c}{c!} \right) \left( p_w \sum_{m=c}^{\infty} C(m)(1-p_w)^m \right) \right) dt$$

Considering $R_1(T)$, recalling that $C(m) = H(m)$ for $m > 0$, we have $p_w \sum_{m=c}^{\infty} C(m)(1 - p_w)^m \leq p_w \sum_{m=1}^{\infty} C(m)(1-p_w)^m = -\ln(p_w)$, as shown previously. Also, for any $c$ we have $\int_0^T \frac{e^{-\Delta_w t}(\Delta_w t)^c}{c!} dt = \frac{\Gamma(c+1)-\Gamma(c+1,\Delta_w T)}{\Delta_w c!}$. Therefore,

$$R_1(T) = \int_0^T \left( \sum_{w \in W} \mu_w \sum_{c=0}^{c^*-1} \left( \frac{e^{-\Delta_w t}(\Delta_w t)^c}{c!} \right) \left( p_w \sum_{m=c}^{\infty} C(m)(1-p_w)^m \right) \right) dt$$

$$< - \sum_{w \in W} \mu_w \ln(p_w) \sum_{c=0}^{c^*-1} \frac{\Gamma(c+1) - \Gamma(c+1, \Delta_w T)}{\Delta_w c!}$$

Since $\lim_{T \to \infty} \frac{\Gamma(c+1,\Delta_w T)}{T} = 0$, $R_1(T)$ is therefore upper-bounded by a finite sum of terms that all go to 0 when divided by $T$ as $T \to \infty$. Since $R_1(T) \geq 0$ also holds, we have $\lim_{T \to \infty} \frac{R_1(T)}{T} = 0$.

Considering $R_2(T)$, we use our definition of $c^*$ to write

$$R_2(T) = \int_0^T \left( \sum_{w \in W} \mu_w \sum_{c=c^*}^{\infty} \left( \frac{e^{-\Delta_w t}(\Delta_w t)^c}{c!} \right) \left( p_w \sum_{m=c}^{\infty} C(m)(1-p_w)^m \right) \right) dt$$

$$< \int_0^T \left( \sum_{w \in W} \mu_w \sum_{c=c^*}^{\infty} \left( \frac{e^{-\Delta_w t}(\Delta_w t)^c}{c!} \right) \left( \frac{1}{c} \right) \right) dt = \int_0^T \left( \sum_{w \in W} \mu_w \sum_{c=c^*}^{\infty} \left( \frac{e^{-\Delta_w t}(\Delta_w t)^c}{(c+1)!} \right) \right) dt$$

$$\leq \int_0^T \left( \sum_{w \in W} \mu_w \sum_{c=0}^{\infty} \left( \frac{e^{-\Delta_w t}(\Delta_w t)^c}{(c+1)!} \right) \right) dt$$

$$= \int_0^T \left( \sum_{w \in W} \mu_w \left( \frac{1 - e^{-\Delta_w t}}{\Delta_w t} \right) \right) dt$$

$$= \sum_{w \in W} \mu_w \frac{\ln(\Delta_w T) + \Gamma(0, \Delta_w T) + \gamma}{\Delta_w},$$

where $\gamma$ is the Euler-Mascheroni constant. Since $\lim_{T \to \infty} \frac{\Gamma(0,\Delta_w T)}{T} = 0$, $R_2(T)$ is therefore upper-bounded by a finite sum of terms that all go to 0 when divided by $T$ as $T \to \infty$. Since $R_2(T) \geq 0$ also holds, we have $\lim_{T \to \infty} \frac{R_2(T)}{T} = 0$. Thus, we have

$$\lim_{T \to \infty} \frac{R(T)}{T} = \lim_{T \to \infty} \frac{R_1(T) + R_2(T)}{T} = \lim_{T \to \infty} \frac{R_1(T)}{T} + \lim_{T \to \infty} \frac{R_2(T)}{T} = 0$$

**Consider** $F(T)$. Observe that for a suitably chosen constant $s > 0$, $sR(T) > F(T)$. Therefore, since $\lim_{T \to \infty} \frac{R(T)}{T} = 0$, $\lim_{T \to \infty} \frac{F(T)}{T} = 0$ too.

We have thus shown that

$$J^{\pi} = \lim_{T \to \infty} \frac{G(T) - R(t) + F(T)}{T} = \lim_{T \to \infty} \frac{G(T)}{T} - \lim_{T \to \infty} \frac{R(T)}{T} + \lim_{T \to \infty} \frac{F(T)}{T} = -\sum_{w \in W} \mu_w \ln(p_w)$$

■

**PROOF OF PROPOSITION 5.** Since under any $\vec{p} \geq 0$ crawl rates $\vec{\rho}$ are related to crawl probabilities via $\rho_w = p_w \Delta_w$, to apply the method of Lagrange multipliers to the relaxation of Problem 2 that takes into account only the bandwidth constraint we set $f(\vec{p}) = \overline{J}^{\pi} = \sum_{w \in W^o} \mu_w \ln(p_w)$ and $g(\vec{\rho}) = \sum_{w \in W^o} p_w \Delta_w - R$. We need to solve

$$\begin{cases} \nabla f(\vec{p}) = \lambda \nabla g(\vec{p}) \\ g(\vec{p}) = 0. \end{cases}$$

For any $w \in W^o$, we have $\frac{\partial g}{\partial p_w} = \Delta_w$ and $\frac{\partial f}{\partial p_w} = \frac{\mu_w}{p_w}$, so the above system of equations turns into

$$\begin{cases} \frac{\mu_w}{p_w} = \lambda \Delta_w, & \text{for all } w \in W^o \\ \sum_{w \in W^o} p_w \Delta_w = R \end{cases}$$

and therefore

$$\begin{cases} p_w = \frac{R \mu_w}{\Delta_w \sum_{w \in W^o} \mu_w} \text{for all } w \in W^o \\ \lambda = \frac{\sum_{w \in W^o} \mu_w}{R} \end{cases}$$

This is the only solution yielded by the method of Lagrange multipliers, so it is the unique maximizer of the relaxation. ■

**PROOF OF PROPOSITION 6.**

To establish LAMBDACRAWL-COMPLOBS's correctness, we first prove the following lemma, which establishes that any source $w$ that violates its $p_w \leq 1$ constraint in any iteration of LAMBDACRAWL-COMPLOBS must have $p_w^* = 1$:

**Lemma 1.** *Let $\vec{p}^*$ be the maximizer of Problem 2, and let $\vec{p}'^*$ be the maximizer of the relaxation Problem 2 with the same inputs but with inequality constraints ignored. Then any source $w$ that has $\vec{p}'^*$ has $p_w'^* > 1$, violating its inequality constraint, necessarily has $p_w^* = 1$.*

*Proof.* For convenience, we rewrite Problem 2 as an equivalent problem of maximizing $\overline{J}_{mod} = \sum_{w \in W^o} \mu_w \ln(\rho_w)$ under the constraints $\sum_{w \in W^o} \rho_w = R$ and $\rho_w \leq \Delta_w$ for every $w \in W^o$. Consider its relaxation with $\rho_w \leq \Delta_w$ for every $w \in W^o$ ignored. By the equivalent of Proposition 5 for this reformulation, $\vec{\rho}'^*$ is unique and must have $p_w'^* = \rho_w'^* / \Delta_w$ for each source $w$. The Lagrangian of $\mathcal{L}(\vec{\rho}, \lambda_0) := \overline{J}_{mod}(\vec{\rho}) - \lambda_0 (\sum_{w \in W^o} \rho_w - R)$ must have $\nabla \mathcal{L}(\vec{\rho}'^*, \lambda_0'^*) = 0$ for the optimal solution $(\vec{\rho}'^*, \lambda_0'^*)$ (which, again, encodes the optimal $p_w'^*$ for the relaxation of the original formulation of Problem 2 via $p_w'^* = \rho_w'^* / \Delta_w$). From this we see that $\frac{\partial}{\partial \rho_u} \overline{J}_{mod} \upharpoonright_{\rho_u'^*} = \lambda_0'^* = \frac{\partial}{\partial \rho_w} \overline{J}_{mod} \upharpoonright_{\rho_w'^*}$, for any sources $u, w \in W^o$. Since $\frac{\partial}{\partial \rho_w} \overline{J}_{mod} = \frac{\mu_w}{\rho_w}$, this implies

$$\frac{\mu_u}{\rho_u'^*} = \frac{\mu_w}{\rho_w'^*} \quad \text{for all } u, w \in W^o. \tag{17}$$

Consider the slack-variable formulation of Problem 2 with slack variables $\{q_w\}_{w \in W^o}$. Inequality constraints in this formulation turn into $\rho_w = \Delta_w - q_w$ for $q_w \geq 0$. In this formulation, we now have

$$\mathcal{L}(\vec{\rho}, \lambda_0, \lambda_1, \ldots, \lambda_{|W^o|}) := \mathcal{L}(\vec{\rho}, \lambda_0) - \sum_{w \in W^o} \lambda_w(\rho_w - \Delta_w) \,.$$

where, by the Karush-Kuhn-Tucker conditions, $\lambda_w \geq 0$ for all $w$. By complementary slackness, $\lambda_w^* = 0$ for every $w \in W^o$ such that $q_w > 0$, i.e., for every $w$ that does *not* activate its inequality constraint, under the optimal solution $(\vec{\rho}^*, \lambda_0^*, \lambda_w^*, \ldots, \lambda_{|W^o|}^*)$. This implies that $\frac{\partial}{\partial \rho_u} \overline{J}_{mod} \upharpoonright_{\rho_u^*}$ $-\lambda_u^* = \lambda_0^* = \frac{\partial}{\partial \rho_w} \overline{J}_{mod} \upharpoonright_{\rho_w^*} -\lambda_w^*$, for any sources $u, w \in W^o$, i.e.,

$$\frac{\mu_u}{\rho_u'^*} - \lambda_u^* = \lambda_0^* = \frac{\mu_w}{\rho_w'^*} - \lambda_w^* \quad \text{for all } u, w \in W^o \,. \tag{18}$$

Now, suppose for contradiction that there is a source $u \in W^o$ that has $p_u'^* > 1$ but $p_u^* < 1$, implying that $\rho_u'^* > \Delta_u$ but $\rho_u^* < \Delta_u$. This, in turn, implies that (a) $\rho_u'^* > \rho_u^*$ and (b) $\lambda_u^* = 0$, since $u$ doesn't activate its inequality constraint under $\vec{\rho}^*$ (and hence under $\vec{p}^*$). Then, since $\sum_{v \in W^o} \rho_v^* = \sum_{v \in W^o} \rho_v'^* = R$, there must also exist some other source $w \neq u$ such that $\rho_w'^* < \rho_w^*$. For this source, $\lambda_w^* \geq 0$, so $\lambda_w^* \geq \lambda_u^*$.

Recall that $\overline{J}_{mod}$ is strictly concave, and its partial derivatives $\frac{\mu_v}{\rho_v}$ are monotone decreasing in every non-negative $\rho_v$. Together with $\rho_u'^* > \rho_u^*$, $\rho_w'^* < \rho_w^*$, $\lambda_w^* \geq \lambda_u^*$, and Equation 18, this implies

$$\frac{\mu_u}{\rho_u'^*} < \frac{\mu_u}{\rho_u^*} \leq \frac{\mu_w}{\rho_w^*} < \frac{\mu_w}{\rho_w'^*}$$

But this contradicts Equation 17, completing the proof of the lemma. $\blacksquare$

The optimality of LAMBDACRAWL-COMPLOBS now follows by induction. Its every iteration except the last one identifies at least one constraint that is active under $\vec{p}^*$, by the above lemma, and thereby assigns an optimal $p_w^*$ to some sources, leaving optimal crawl probabilities for others to be found in subsequent iterations. The solution for the sources remaining in the final iteration, which does not violate any inequality constraints, is optimal by Proposition 5. Therefore, LAMBDACRAWL-COMPLOBS arrives at the optimal solution, and that solution is unique because Problem 2's maximization objective is concave as a sum of concave functions, and the optimization region is convex.

We note that the proof so far is similar to the proof of Lemma 3.3 from [21] for a different concave function $F$ under constraints of the form $x_1 + \ldots + x_k \leq c_k$, where $x_1, \ldots, x_k$ is a subset of $F$'s variables and $c_k$ is a constraint.

Since in each iteration LAMBDACRAWL-COMPLOBS removes at least one source from further consideration, it makes at most $|W^o|$ iterations. In each iteration it applies Proposition 5, which takes $O(|W^o|)$ time, yielding the overall time complexity of $O(|W^o|^2)$. $\blacksquare$

**PROOF OF PROPOSITION 7.** See the paper. $\blacksquare$

**PROOF OF PROPOSITION 8.** LAMBDALEARNANDCRAWL starts with strictly positive finite estimates $\vec{\Delta}_0$ of change rates. Since LAMBDACRAWL, which LAMBDALEARNANDCRAWL uses for determining crawl rates for the next epoch, is optimal to any desired precision (Proposition 7), it follows from Corollary 1 that it returns positive $\vec{\rho}_1^*, \vec{p}_1^* > 0$, and $0 < \vec{\mu}, R < \infty$ guarantees that these crawl rates are also finite. In subsequent iterations, $\vec{\Delta}_n$ are estimated using Equations 10 and 11 with smoothing terms (lines 11 and 13 of Algorithm 4), and the smoothing terms ensure that the change rate estimates are finite and bounded away from 0: $0 < \delta_{min} \leq \vec{\Delta}_n < \infty$, where $\delta_{min}$ is implied by the aforementioned estimators and specific smoothing term values. This, along with finite positive $\vec{\mu}$

and $R$, ensures that $0 < \vec{\hat{\rho}}^*_{n+1}, \vec{\hat{p}}^*_{n+1} < \delta_{max}$. Hence, by induction, no source is ever starved, and no source is crawled infinitely frequently. This ensures, together with consistency of estimators from Equations 10 and 11, that if at least the last iteration $N_{epoch}$ uses the entire observation history, i.e. $S(N_{epoch}) = length(obs\_hist)$, then change rate estimates converge to the true change rates in probability: $\text{plim}_{N_{epochs} \to \infty} \vec{\hat{\Delta}}_{N_{epochs}} = \vec{\Delta}$, as long as $\vec{\Delta}$ doesn't change with time. Optimality of LAMBDACRAWL then implies probabilistic convergence of $(\vec{\hat{\rho}}^*_{N_{epochs}}, \vec{\hat{p}}^*_{N_{epochs}})$ to $(\vec{\hat{\rho}}^*, \vec{\hat{p}}^*)$ as well. ∎

**PROOF OF PROPOSITION 9.** According to Equation system 5, the parameter vector $\vec{\rho}$ of the optimal $\pi^* \in \Pi^-$ that minimizes the expected harmonic penalty in the absence of remote change observations (Problem 1) satisfies

$$\begin{cases} \rho_w = \frac{-\Delta_w + \sqrt{\Delta_w^2 + \frac{4\mu_w \Delta_w}{\lambda}}}{2}, & \text{for all } w \in W^- \\ \sum_{w \in W^-} \rho_w = R \end{cases}$$

If $\frac{\mu_w}{\Delta_w} = c$ for all $w \in W^-$, $c > 0$, then we can express $\Delta_w = c'\mu_w$ where $c' = 1/c$ and plug it into the above equations to get

$$\begin{aligned}
\rho_w &= \frac{-\Delta_w + \sqrt{\Delta_w^2 + \frac{4\mu_w \Delta_w}{\lambda}}}{2} \\
&= \frac{-c'\mu_w + \sqrt{(c'\mu_w)^2 + \frac{4c'\mu_w^2}{\lambda}}}{2} \\
&= \frac{-c'\mu_w + \mu_w\sqrt{\frac{c'^2\lambda + 4c'}{\lambda}}}{2} \\
&= \mu_w \left( \frac{-c' + \sqrt{\frac{c'^2\lambda + 4c'}{\lambda}}}{2} \right) \quad \text{for all } w \in W^-
\end{aligned} \tag{19}$$

Plugging this into the remaining equation from the above system, $\sum_{w \in W^-} \rho_w = R$, we get

$$\sum_{w \in W^-} \mu_w \left( \frac{-c' + \sqrt{\frac{c'^2\lambda + 4c'}{\lambda}}}{2} \right) = R \implies$$

$$\frac{-c' + \sqrt{\frac{c'^2\lambda + 4c'}{\lambda}}}{2} = \frac{R}{\sum_{w \in W^-} \mu_w} \implies$$

$$\frac{c'^2\lambda + 4c'}{\lambda} = \left( \frac{2R}{\sum_{w \in W^-} \mu_w} + c' \right)^2 \implies$$

$$\lambda \left( \frac{2R}{\sum_{w \in W^-} \mu_w} + c' \right)^2 - c'^2\lambda = 4c' \implies$$

$$\lambda = \frac{4c'}{\left( \frac{2R}{\sum_{w \in W^-} \mu_w} + c' \right)^2 - c'^2}$$

Plugging this and $\Delta_w = c'\mu_w$ back into Equations 19, we get for all $w \in W^-$

$$\rho_w = \mu_w \left( \frac{-c' + \sqrt{\frac{c'^2\left(\frac{4c'}{\left(\frac{2R}{\sum_{w\in W^-}\mu_w}+c'\right)^2-c'^2}\right)+4c'}{\frac{4c'}{\left(\frac{2R}{\sum_{w\in W^-}\mu_w}+c'\right)^2-c'^2}}}}{2} \right)$$

$$= \mu_w \left( \frac{-c' + \sqrt{\frac{4c'\left(\left(\frac{c'^2}{\left(\frac{2R}{\sum_{w\in W^-}\mu_w}+c'\right)^2-c'^2}\right)+1\right)}{\frac{4c'}{\left(\frac{2R}{\sum_{w\in W^-}\mu_w}+c'\right)^2-c'^2}}}}{2} \right)$$

$$= \mu_w \left( \frac{-c' + \sqrt{\frac{\left(\frac{2R}{\sum_{w\in W^-}\mu_w}+c'\right)^2\left(\left(\frac{2R}{\sum_{w\in W^-}\mu_w}+c'\right)^2-c'^2\right)}{\left(\frac{2R}{\sum_{w\in W^-}\mu_w}+c'\right)^2-c'^2}}}{2} \right)$$

$$= \mu_w \left( -c' + \frac{R}{\sum_{w\in W^-}\mu_w} + c' \right)$$

$$= \frac{\mu_w R}{\sum_{w\in W^-}\mu_w}$$

∎