[Reviews · NeurIPS 2019]

Reviewer 1



The paper formulates the problem of optimizing a strategy for crawling remote contents to track their changes as an optimization problem called the freshness crawl scheduling problem. This problem is an obviously important problem in applications like Internet search engine, and the presented formulation seems to give a practical solution to those applications. The paper presents an algorithm for solving the freshness crawl scheduling problem to optimality, assuming that the contents change rates are known. The idea behind the algorithm is based on the deep understanding of statistics and continuous optimization, and it seems to me that the contribution is solid (although I could not very all the technical details). For the case where the contents change rates are not known, a reinforcement learning algorithm is presented. Basically it estimates the rates as their MLEs. As a learning algorithm, the approach is not non-trivial, but I think it is reasonable. The convergence of the algorithm is guaranteed, and it is also valuable. The authors provided experimental results as well. They crawled 18,532,326 URLs daily over 14 weeks. It shows that the presented algorithm scales. This is an important feature because the web crawling requires processing a large number of sites. To conclude, the paper presents reasonable formulations and algorithms for the important practical problem. The technical contributions are solid from aspects of both theoretical optimization study and practical engineering work. The paper is written carefully and is easy to follow. I slightly worry that this paper is not very suitable for presentation at machine learning conferences (to my opinion, this paper is more suitable for data mining or web conferences), but I think that the paper is a good contribution even for Neurips. Update: After reading the author's rebuttal, I still maintain my score.

Reviewer 2



The paper presents the problem meticulously by defining the different components affecting the freshness of the crawler. In order to maximize the freshness, the problem is modeled as minimizing the cost of tracking defined through a harmonic staleness penalty function, where the authors elaborate on the choice of this harmonic function with sound reasons. The cost function is optimized according to how the change is observed in the source, whether complete or incomplete. The authors present two algorithms for each change type, supplemented with prepositions and their proof. These algorithms are cast into a model-based Reinforcement learning problem that tries to automatically learn the change/update rate of each source. The paper lacks organization and proper presentation. It relies on several details that are only presented in the supplement (extra ~15 pages). As is, the paper is only a problem description and presentation of a proposed approach. Hence, it is missing a conclusion, description of the dataset, details of the experiment and proper analysis.

Reviewer 3



In this article, the authors propose methodology for scheduling a web crawler. They frame their scheduler as characterized by the solution to an optimization problem (that characterizes the cost of staleness). They show how to determine parameters of the scheduler when parameters of the environment are known. In addition, they extend their algorithm to learn parameters of the environment, and optimize against them when they are unknown. The authors additionally provide optimality guarantees for their methodology. The article is generally very clear. The introduction and problem formulation do a great job of clearly presenting the problem. Near the end of the paper, I struggled with the exposition a bit more in the reinforcement learning section, though perhaps this section is necessarily dense (given space constraints of the paper, and the additional complexity of the section). It wasn't clear to me in the case of complete change observations if the method proposed was optimal, or if was only that the optimal p_w were selected for strategies within that class. Some clarification here would be great. It is noted that the first two problems are convex. The algorithms given solve each problem in O(|W|^2) [and it is stated that practically the run time is closer to linear]. It would be worth noting the run-time of a general interior point solver --- I imagine this is O(|W|^3). The empirical experiments seemed well thought out, and fairly illustrative.

[Author Response · NeurIPS 2019]

Dear reviewers,

Thank you for your feedback. Here are our responses to your questions and comments:

**Reviewer 1:** Regarding Web and data mining conferences, we agree that this work is relevant to them as well. However, we believe that due to stronger emphasis on optimization and ML rather than, say, on the empirical details of web page change modelling, it will be more appreciated by the NeurIPS audience. Indeed, of the three recent works most related to ours topic-wise (citations [2], [30], and [32]), two were published in ICML and NeurIPS.

**Reviewer 2:** To answer your question about domain-level modeling of change rates: absolutely! This can be and is done in practice. In the same vein, it is common to do it at the site level. The easiest way to incorporate these change rate correlations into our approach is to use, e.g., a historical change rate average computed over pages from a given domain as the initial change rate "guess" for new pages from this domain, with appropriately chosen pseudocounts. (In Algorithm 4 the pseudocounts are 0.5, but for informed change rate estimates they should be higher.) This won't affect our RL algorithm's theoretical guarantees, but will certainly improve its empirical convergence rate.

Concerning the scarce description of the dataset, experiments, and their analysis, we indeed had to make some hard decisions in this area to conform to the initial submission's 8-page limit. However, if the paper is accepted, we will use the extra 9th page to accommodate a shortened version of the evaluation metrics' and benchmark algorithms' description from the Supplement (Sections 8.2 and 8.3). We will also move some of analysis from the Supplement's Sections 8.4, 8.5, and 8.6's to that 9th page — a condensed version of most of their content is in Figure 1, 2, and 3's captions in the main paper already. Last but not least, we will move a few details from the Supplement's Section 8.1 (*Dataset, Implementation, Hardware*) into the paper too. Much of Section 8.1's content will be in the writeup accompanying the dataset itself though, which we will make public prior to NeurIPS. These changes will make the paper's final version more self-contained than it currently is, and even leave space for a short conclusion.

**Reviewer 3:** To clarify the claim about the optimality of our complete-observation algorithm: currently, we can only claim that the policy it computes is optimal within its policy class, defined in Equation 6. We *believe* that de-randomizing this class's best policy in some way should yield a globally optimal policy, but so far don't have a proof.

Mentioning interior-point methods is an excellent suggestion, we'll do that. Their complexity varies depending on their assumptions about the objective and optimization region. Table 1.1 in http://sbubeck.com/Bubeck15.pdf summarizes relevant results for interior-point as well as other convex optimization algorithms. For those whose single iteration is equivalent to an iteration of Newton's method, the per-iteration complexity would indeed be around $O(|W|^3)$.

Best regards,

–The authors of submission #315.

[Meta-Review · NeurIPS 2019]

This paper presents an RL-approach for optimizing web crawling strategy by modeling freshness of the remote content. Reviewers were unanimously in favor of accepting the paper, appreciating the formulation of the problem and the extent of the scale of experiments.